# Dynamic characteristics of scraper conveyor chain drive system under the impact condition of lump coal

**Shoubo Jiang, Jinwang Lv, Qingliang Zeng[ID]\*, Qiang Zhang, Yuqi Zhang, Wei Qu, Jiexu Cui**

College of Mechanical and Electronic Engineering, Shandong University of Science and Technology, Qingdao, China

\* qlzeng@163.com

**Data Availability Statement:** All relevant data are within the manuscript and its Supporting Information files.

## Abstract

Scraper conveyor is the most important transportation equipment in the comprehensive mining equipment, and the chain drive system is its core subsystem, its dynamic characteristics will significantly affect the efficiency of coal transportation in the comprehensive mining face. In this paper, the dynamic characteristics of chain drive system when impacted by falling coal are investigated by means of test. The impact test bench of scraper conveyor was set up to analyze the effects of chain speed, impact height and impact load mass on the dynamic characteristics of the chain drive system of scraper conveyor under the working conditions of unloaded and loaded. The results show that the longitudinal vibration of the scraper conveyor is most obvious when it is impacted by the falling coal, and the chain speed, impact height and impact load mass of the scraper conveyor all play an excitation role on the vibration of the chain drive system, and the vibration of the chain ring is the most intense in the chain drive system, and the loaded coal pile conveyed on the scraper conveyor plays an inhibiting role on the vibration of the chain drive system. This study can help to identify the location where the scraper conveyor fails first in the impact condition, so as to provide a basis for its structural design and improvement, which is of great significance for the stable operation and structural optimization of the scraper conveyor.

## 1 Introduction

Long-wall integrated mining technology is the most widely used technology in coal mining technology at present, and it is also the main direction of coal mining technology development [1]. Long-wall comprehensive mining face mainly consists of coal mining machine, scraper conveyor, hydraulic support and other auxiliary equipment, and scraper conveyor is the most important transportation equipment in comprehensive mining equipment [2, 3]. The working environment of scraper conveyor is more complicated, in the process of coal mining there are often large pieces of coal rock collapsed to the scraper conveyor, the chain drive system of the scraper conveyor caused by the obvious impact, which seriously affects the dynamics characteristics of the scraper conveyor, and in serious cases, it will even make the scraper conveyor

**Funding:** This study was supported by the National Natural Science Foundation of China (52204143, 52234005), Qingdao Postdoctoral Funding Project (QDBSH20230102027), and the Shandong Provincial Key Laboratory of Mining Machinery Engineering Open Fund (University-Enterprise Joint Funding Project, 2022KLMM307), and the China Postdoctoral Science Foundation(Grant No. 2023M742140).

**Competing interests:** The authors have declared that no competing interests exist.

broken chain, chain jamming and other failures [4, 5]. Therefore, it is necessary to carry out experimental research on the dynamic characteristics of scraper conveyor under the action of impact load, which can provide theoretical guidance for the structural design and optimization of scraper conveyor, and is of great significance for the safe and stable operation of scraper conveyor.

At present, scholars have carried out a lot of research work on the dynamic characteristics of scraper conveyor under various working conditions, mainly through theoretical research, simulation analysis and experimental research on the dynamic characteristics of the chain drive system of scraper conveyor. Ma et al. [6] studied the wear of the central groove of the scraper conveyor by establishing a coupled simulation model of the central groove wear based on DEM-MBD, and used the model to analyze the stress and motion state of the scraper, and then studied the wear of the central groove, and the results of the study provided a simulation basis for the study of the wear resistance of the central groove. Lu et al. [7] established a coupled simulation model of torsional and longitudinal vibration of the chain drive system of scraper conveyor based on the point-by-point tension method and the Kelvin-Voigt model, and carried out numerical simulations, and verified the correctness of the model through the comparative analysis with the experiments. Ren et al. [8] established a coupled analytical model of scraper conveyor based on multi-body dynamics theory and discrete element method, and simulated the dynamic characteristics of the scraper conveyor transporting different load conditions, and the results showed that the transverse vibration and longitudinal vibration of the chain were positively and negatively correlated with its coal conveying capacity, respectively. Yao et al. [9] obtained the simulation data of normal contact energy and impact damage depth of the middle groove of scraper conveyor based on orthogonal matrix analysis and discrete element method, and established the contact energy matrix and impact damage depth matrix to get the optimal working condition combination of scraper conveyor. Hao et al. [10] designed a new type of scraper, which can convert sliding friction into rolling friction, and used SolidWorks Simulation to carry out finite element analysis on its three-dimensional model, and the results show that the new type of scraper helps to extend the service life of the scraper conveyor. Wojnar et al. [11] experimentally investigated the vibration of the reducer of a scraper conveyor by means of determining the position of the gear teeth on the reducer of the scraper conveyor, which in turn analyzed the dynamics of the scraper conveyor. Zhang et al. [12] designed a well-shaped scraper reinforcement through SolidWorks modeling software and Ansys software, which significantly improved the operational reliability of the scraper and increased the service life of the scraper. Lu et al. [7] established a torsional and longitudinal coupling analysis model of scraper conveyor chain drive system based on the point-by-point stretching method in order to study the mechanical characteristics of scraper conveyor under loaded condition, and carried out numerical simulation and analysis, and finally verified the reasonableness of the model through tests. Zhang et al. [13] established a multi-body dynamics model of the chain drive system of a scraper conveyor and verified the reliability of the dynamics model through fixed-point strain measurement experiments. Zhang et al. [14] proposed a scraper chain tension prediction method based on sensor information fusion for chain drive system of scraper conveyor by combining the improved D-S evidence theory, which was proved to be significantly better than other conventional methods in terms of the correctness of the predicted parameters through experiments.

Shprekher et al. [15, 16] used a multi-mass system to establish a numerical simulation model of the chain drive system of a scraper conveyor and carried out simulation of the no-load and full-load startup process under multi-source drive, and the simulation results show that the simulation speed of the simulation model established under the multi-mass system is faster than that of the single-mass model, and the error is smaller. Murphy et al. [17] proposed

a method to study the load variation of a scraper conveyor and analyzed the dynamic characteristics of the scraper conveyor during load variation. Ma et al. [18] studied the operational wear of the central groove of a scraper conveyor, and simulated the wear of the central groove of a scraper conveyor by establishing a coupled wear analysis model of the central groove of a scraper conveyor. Xie et al. [19] investigated the torsional-sway vibration characteristics of the chain drive system under the unilateral stuck-chain condition, and analyzed the simulated application of the stuck-chain load and the causes of the stuck-chain failure. Jiang et al. [20] constructed a virtual prototype model of scraper conveyor, and analyzed the dynamic characteristics of the scraper conveyor in the impact conditions, the emergence of broken and jammed chains and other fault conditions by simulation, which provided a reference for the structural optimization design of the scraper conveyor.

Anna Grincova et al. [21] studied the damage characteristics of scraper conveyors under impact conditions and investigated the impact resistance characteristics and service life of different scraper conveyors under impact conditions by means of experiments. Jiang et al. [22, 23] investigated the effects of impact position, impact load and impact height on the chain drive system of scraper conveyor by means of joint simulation of discrete element and multi-body dynamics, and simulated several most common coal conveying conditions of scraper conveyor by building a test bed of the chain drive system of the scraper conveyor, and comparatively analyzed the dynamic characteristics of the chain ring by collecting acceleration signals. Si et al. [24] analyzed the causes of normal impact of vertical chain drive system of scraper conveyor, calculated the operating loads of scraper conveyor under different impact forces, and analyzed the impact loads of scraper conveyor under various impact conditions by establishing the impact simulation model of scraper conveyor.

Through analyzing the research of the above scholars, it is found that most of the research on the dynamic characteristics of scraper conveyor under normal coal conveying condition and impact condition analyzes the components such as its middle groove or chain ring independently, and does not analyze the components of each part comparatively. And when studying the dynamic characteristics of the scraper conveyor under impact conditions, the difference in characteristics between the unloaded and loaded conditions was not analyzed, at the same time, due to the simulation and analysis of the scraper conveyor, not only need to define a large number of relevant parameters, but also can not simulate the real situation of coal conveying conditions, and the real situation is more different. Therefore, this paper adopts the way of constructing the impact test bench of scraper conveyor to study the influence of different chain speeds, different impact heights and different impact load qualities on the dynamic characteristics of the chain drive system under the two working conditions of unloaded and loaded, so as to obtain the changes of the dynamic characteristics of the scraper conveyor when it is subjected to impact under different working conditions. This study can provide a theoretical basis for the design and improvement of the scraper conveyor and help to predict and prevent potential failures, which is of great significance for the stable operation and structural optimization of the scraper conveyor.

## 2 Theoretical basis

Since this study involves the impact problem of coal block and scraper conveyor, and impact dynamics is the theory that mainly studies the deformation, motion or destruction law of objects or materials when they are subjected to the action of variable impact load within a short period of time, this paper is based on the theory related to impact dynamics to study the change of dynamics characteristics of scraper conveyor when it is subjected to the impact of falling coal.

When investigating the dynamic characteristics of scraper conveyors under impact loads, it is necessary to simplify the impact loads to concentrated instantaneous loads and to disregard the effect of friction on the impact results during the impact process. According to the basic principles of impact dynamics, the impact problem can be simplified as an external force F exerting an impact force on an object or system with N mass points. Based on the basic principles of impact dynamics, the process of deriving the fundamental equations of the impact problem is as follows:

Let the mass of the $i$th mass point in the system be $m_i$ and the bit vector relative to the inertial coordinate system be $r_i$, then the following equation can be obtained by the center of mass theorem [25]:

$$\frac{dI}{dt} = P(t) \tag{1}$$

$$I = \sum_{i=1}^{N} m_i r_i = mr_c \tag{2}$$

$$m = \sum_{i=1}^{N} m_i \tag{3}$$

Where I is the system momentum; m is the system mass; and r is the bit vector of the center of mass relative to the inertial coordinate system.

Integrating Eq (1) over the time interval $[t_1, t_2]$ gives:

$$I(t_2) - I(t_1) = P^* \tag{4}$$

$$P^* = \int_{t_1}^{t_2} P(t)dt \tag{5}$$

Where P* is the external impulse, where Eq (5) is the integral expression of the momentum theorem, which means that the total momentum of the system of particles is equal to the external impulse of the system of particles in a certain time period.

Furthermore, if the structural system contains rigid bodies, applying the momentum moment theorem to the center of mass has:

$$\frac{dH}{dt} = G(t) \tag{6}$$

Where H is the system momentum and G is the total external moment.

Integrating Eq (6) over the time interval $[t_1, t_2]$ gives:

$$H(t_2) - H(t_1) = M^* \tag{7}$$

$$M^* = \int_{t_1}^{t_2} G(t)dt \tag{8}$$

Where M* is the impulse moment; Eq (8) is the integral expression form of the momentum moment theorem, which means: in a certain time period, the change in the impulse moment acting on the system is equal to the change in the momentum moment of the system.

From Eq (4), the change in momentum of the system is:

$$\Delta I = I(t_2) - I(t_1) \tag{9}$$

From Eq (7), the change in the momentum moment of the system is:

$$\Delta H = H(t_2) - H(t_1) \tag{10}$$

From the principle of integration, when the value of the external force F is large and the interval $\Delta t$ is very small, Eqs (4) and (7) can be simplified as:

$$\Delta I = P^* \tag{11}$$

$$\Delta H = M^* \tag{12}$$

The above Eqs (11) and (12) are the basic equations for solving the impact problem.

## 3 Research program

In this paper, the impact test bench of scraper conveyor is set up to simulate the three most common working conditions of scraper conveyor under unloaded and loaded working conditions: variable chain speed, variable height of impact, and variable mass of impact load working conditions. This chapter will mainly introduce the construction of the impact test bench and the design of the test program, the impact test bench used in the scraper conveyor and sensor models are common models in the industry, and the variable range set in the test program for the test bench can simulate the most in line with the actual working conditions of the variable range.

### 3.1 Build test bench

The schematic diagram of the scraper conveyor impact test bed constructed in this paper is shown in Fig 1, which mainly consists of a scraper conveyor, the impact device of the large lump coal, and a data acquisition and analysis system. Among them, the scraper conveyor model is SGD320/17B, the specific parameters are shown in Table 1 below, and the sensor model is BWT901CL acceleration sensor, the specific parameters are shown in Table 2 below.

Fig 2 below shows the schematic diagram of the impact device of the large lump coal, which consists of several parts such as gantry, hand winch, wire rope, impact platform and so on. The test is conducted by manually loading the coal blocks onto the impact platform and remotely controlling the retraction of the electric lock below the impact platform to open the lower bottom plate of the impact platform, so that the coal blocks fall down and impact on the scraper conveyor. This device can adjust the height of the impact platform through the hand winch, and then adjust the height of the impact of the coal block, which can realize the impact of the height of the range of 40cm~220cm, can better simulate the actual working conditions. At the same time, four traveling wheels are set below this device, which can make it move back and forth along the running direction of the scraper conveyor, and then adjust the impact position of the coal block.

### 3.2 Test scheme

The test scheme of this study is shown in Fig 3 above, the acceleration sensors were installed on the chain ring and scraper at the front side of the impact position as well as on the central groove at the impact position, and the acceleration changes at the chain ring, scraper and central groove were detected by the supporting data acquisition system, as shown on the left side

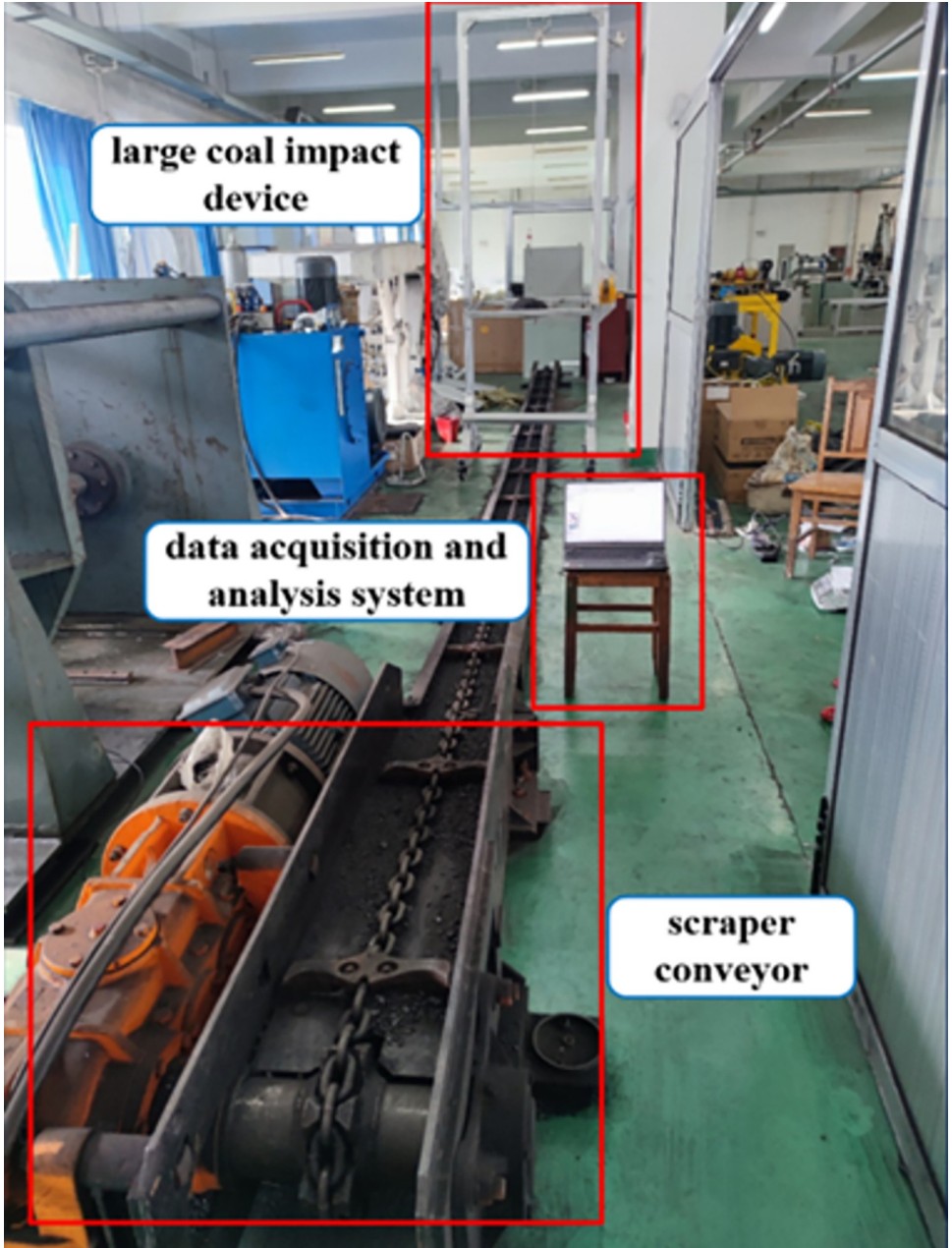

**Fig 1. Scraper conveyor impact test bench.**

of Fig 3 above. Acceleration sensors are installed in the following directions: X direction is the running direction of the scraper chain, Z direction is the longitudinal direction, i.e. the opposite direction of gravity, and Y direction is the transverse direction perpendicular to the X and Z directions.

This test bench is able to simulate two real working conditions: no-load condition and loaded condition. When simulating the loaded condition, the load transported between each two sections of scraper conveyor is fixed at 2kg, which is loaded into the scraper conveyor by manual weighing and manual loading.

**Table 1. Parameters of scraper conveyor.**

| SGD320/17B Scraper Conveyor | |
|---|---|
| Power of variable frequency motors (kW) | 18.5 |
| Rotation speed of the motor (rpm) | 1470 |
| Length of test stand (m) | 12 |
| Dimensions of central groove (L×W×H, mm) | 600×320×156 |
| Rated chain speed (m/s) | 0.59 |
| Rated operating voltage (V) | 380 |
| Reducer ratios | 24.95 |
| Breaking force of round link chains (kN) | >250 |

In this test, coal impacts were carried out at three positions under no load and loaded conditions: at the chain ring in no load condition, at the chain ring at the front side of the loaded coal pile in loaded condition, and at the loaded coal pile in loaded condition.

The impact tests of variable chain speed, variable impact height and variable impact load mass were carried out at the above three positions respectively. In the impact test of variable chain speed, the scraper conveyor chain speed was adjusted by adjusting the frequency of the frequency converter, and the corresponding relationship between the frequency of the frequency converter and the chain speed of the scraper conveyor is shown in Table 3 below. In the impact test of variable height, the height of the impact platform is adjusted by means of a hand winch on the right side of the impact device of the large lump coal, which in turn changes the impact height. In the impact test of variable mass, the mass of the impact load is varied by selecting different masses of coal blocks. The frequency inverter, the bulk coal impact device and bulk coal are shown on the right side of Fig 3 above. The specific values of chain speed, impact height and mass of impact load for the scraper conveyor are shown in Table 4 below.

The specific implementation process of this test was:

1. Place the bulk coal on the impact platform on the impact device of the large lump coal, adjust the hand winch on the impact device of the large lump coal, and pull the impact platform along the guide rail to reach the specified height by the wire rope.

2. Start the scraper conveyor, adjust the chain speed of the scraper conveyor to the specified value through the frequency converter, open the electric control lock under the impact platform through the remote control, and the bulk coal fall down and impact to the specified position of the scraper conveyor by the effect of gravitational acceleration.

3. The scraper conveyor vibrates after being impacted by the falling coal, and the acceleration sensor collects the acceleration data at each position and uploads it to the supporting data acquisition and analysis system for processing and storage.

**Table 2. Parameters of acceleration sensors.**

| BWT901CL Sensor | |
|---|---|
| Rated operating voltage (V) | 3.3~5 |
| Rated operating current (mA) | <40 |
| Acceleration range (m/s$^2$) | ±16g |
| Stability of measurements (m/s$^2$) | 0.01g |
| Frequency of data return (Hz) | 0.1~200 |
| Bluetooth transmission distance (m) | >10 |

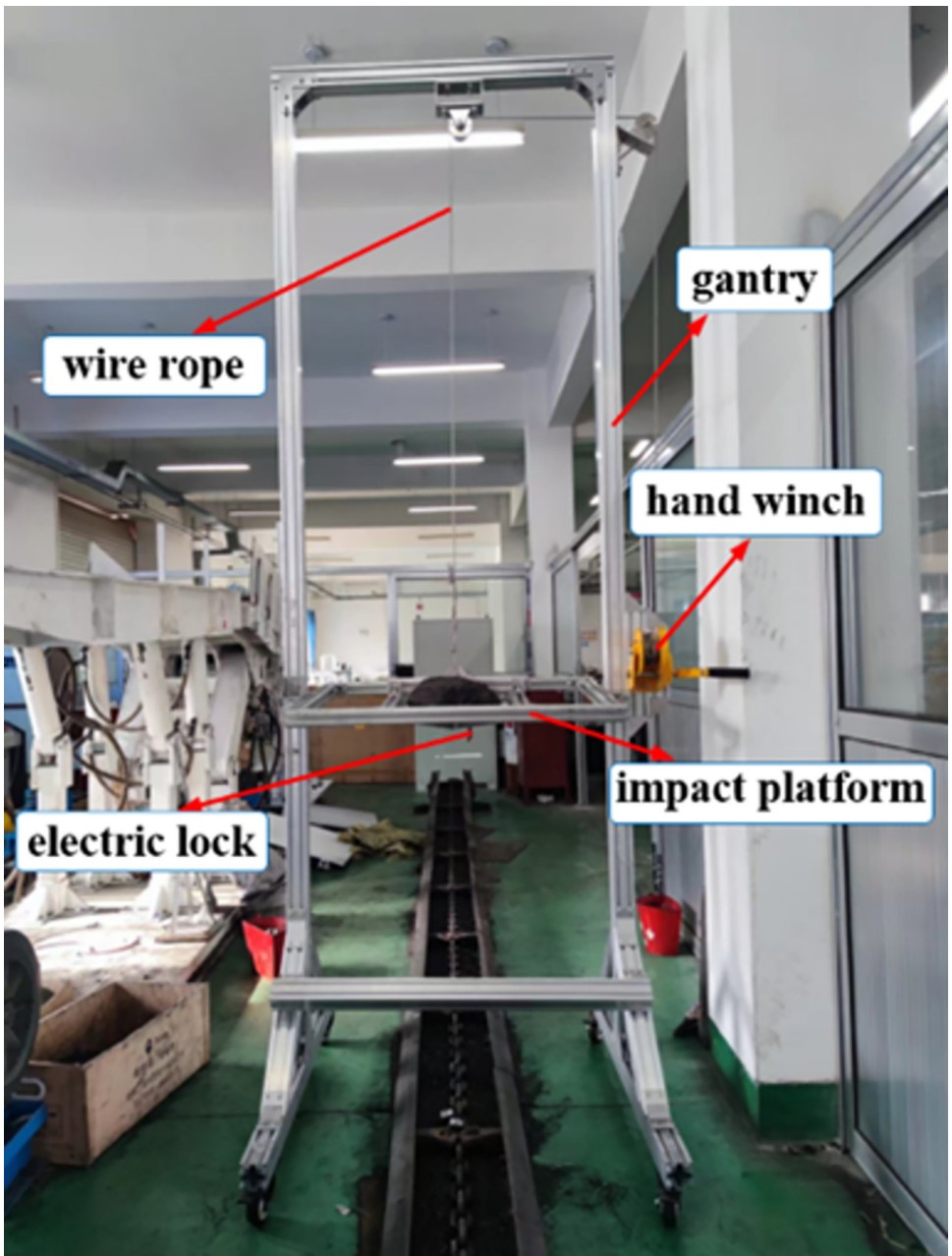

**Fig 2. The impact device of the large lump coal.**

## 4 Test results and analysis

After completing the construction of the impact test bench of the scraper conveyor and the design of the test scheme, the content of this chapter will introduce the acquisition and analysis of the test results, respectively, to carry out the impact test under the no-load condition and under the loaded condition, and to collect the acceleration signals of the chain drive system through the data acquisition and analysis system.

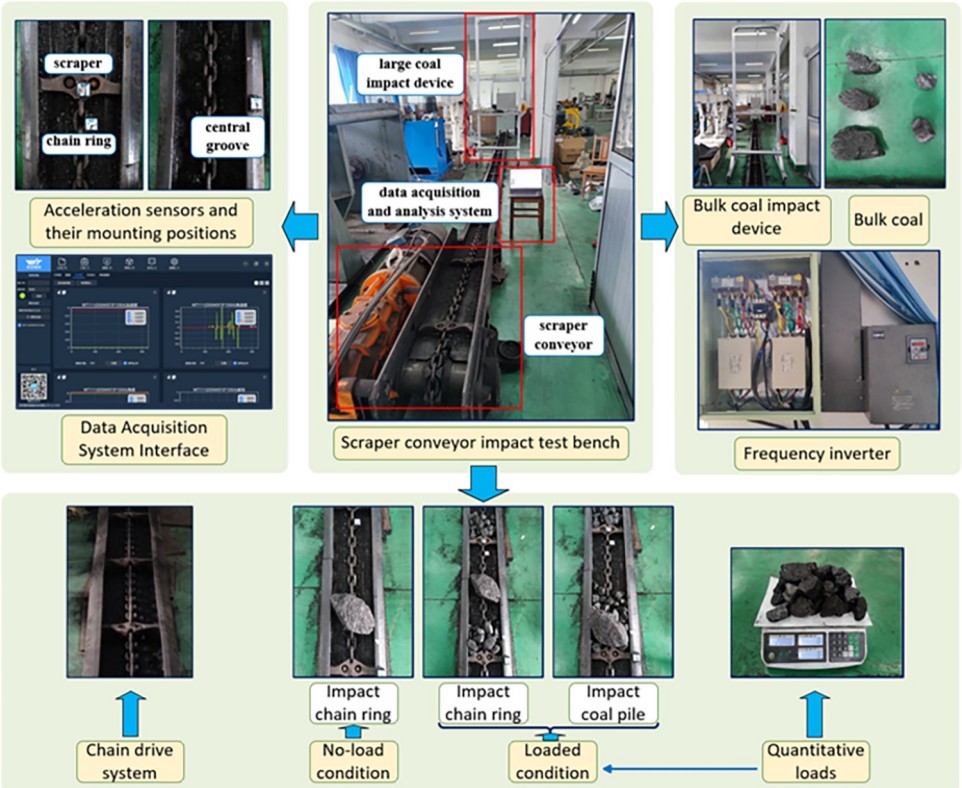

**Fig 3. Test scheme for impact scraper conveyors.**

## 4.1 No-load condition

Based on the test bench of scraper conveyor and impact test scheme described in the previous section, the impact test of the chain drive system under no-load condition is carried out firstly, and the schematic diagram of the coal block impacting the scraper conveyor is shown in Fig 4 below.

**4.1.1 Variable chain speed condition.** In the variable chain speed impact condition, the impact height is taken as 100cm, the impact load mass is taken as 5kg, and the chain speed of the scraper conveyor is changed by adjusting the frequency of the frequency converter. When

**Table 3. Frequency-Chain speed correspondence table.**

| Frequency/(Hz) | Chain speed/(m/s) |
|---|---|
| 4 | 0.047 |
| 6 | 0.071 |
| 8 | 0.094 |
| 10 | 0.118 |
| 12 | 0.141 |
| 14 | 0.165 |
| 16 | 0.189 |
| 18 | 0.212 |
| 20 | 0.236 |
| 22 | 0.259 |

**Table 4. Variables of the impact test.**

| Variables of the impact test | Chain speed/(m/s) | Height of impact/(cm) | Mass of the impact load/(kg) |
|---|---|---|---|
| Values of variables | 0.047 | 40 | 1.5 |
| | 0.071 | 60 | 2.0 |
| | 0.094 | 80 | 2.5 |
| | 0.118 | 100 | 3.0 |
| | 0.141 | 120 | 3.5 |
| | 0.165 | 140 | 4.0 |
| | 0.189 | 160 | 4.5 |
| | 0.212 | 180 | 5.0 |
| | 0.236 | 200 | |
| | 0.259 | 220 | |

the scraper conveyor is impacted under different chain speed conditions, the changes of acceleration in X, Y and Z directions at the three positions of the impacted chain ring, scraper and central groove are collected, as shown in Fig 5 below.

As shown in Fig 5, when the scraper conveyor is running smoothly, the three-way acceleration of the chain drive system is basically zero, and at the moment of impact, the acceleration at the three positions fluctuates drastically, and the greater the chain speed of the scraper conveyor, the greater the acceleration fluctuation at the time of impact.

Calculate the average value and standard deviation of acceleration in X, Y and Z directions at three positions under the two working conditions of minimum chain speed 0.047m/s and maximum chain speed 0.259m/s in this test, as shown in Table 5 below.

As shown in Table 5, with the increase of chain speed from 0.047m/s to 0.259m/s, the average acceleration of chain ring, scraper and central groove at all three places increased significantly, in which the largest fluctuation of acceleration and standard deviation was the chain ring, followed by scraper, and lastly is the central groove. This is due to the fact that in the impact test, the coal block is directly impacted to the chain ring, so the acceleration fluctuation of the chain ring is the largest, secondly, due to the scraper is fixed in the central groove, the latter to a certain extent to play a role in suppressing its vibration, so the acceleration fluctuation is smaller than the chain ring, and finally, due to the central groove for the chain ring and the scraper's carrier and the whole is placed in the ground, the most stable, so the acceleration fluctuation of the smallest.

**4.1.2 Variable height condition.** In the variable height impact condition, the chain speed of the scraper conveyor is set to 0.118m/s, the mass of the impact load is set to 5kg, and the height of the impact load is changed by the hand winch on the impact device of the large lump coal. When the scraper conveyor was impacted under different impact height conditions, the changes of acceleration in X, Y and Z directions at the impacted chain ring, scraper and central groove were collected, as shown in Fig 6 below.

As shown in Fig 6 above, similar to the variable chain speed condition, when the scraper conveyor is impacted by the falling coal, the three-way acceleration of the chain drive system fluctuates drastically, and the higher the impact height of the coal block, the greater the degree of fluctuation of the acceleration.

Statistics of the three-direction maximum acceleration of the chain ring, scraper and central groove at different impact heights were obtained, and the maximum acceleration of the chain drive system in the X, Y and Z directions during the increase of the impact height from 40cm to 220cm is shown in Fig 7 below.

As shown in Fig 7 above, firstly, in the X direction, the maximum acceleration of chain ring increased from 2.188m/s$^2$ to 9.522m/s$^2$, which increased 335.19%; the maximum acceleration

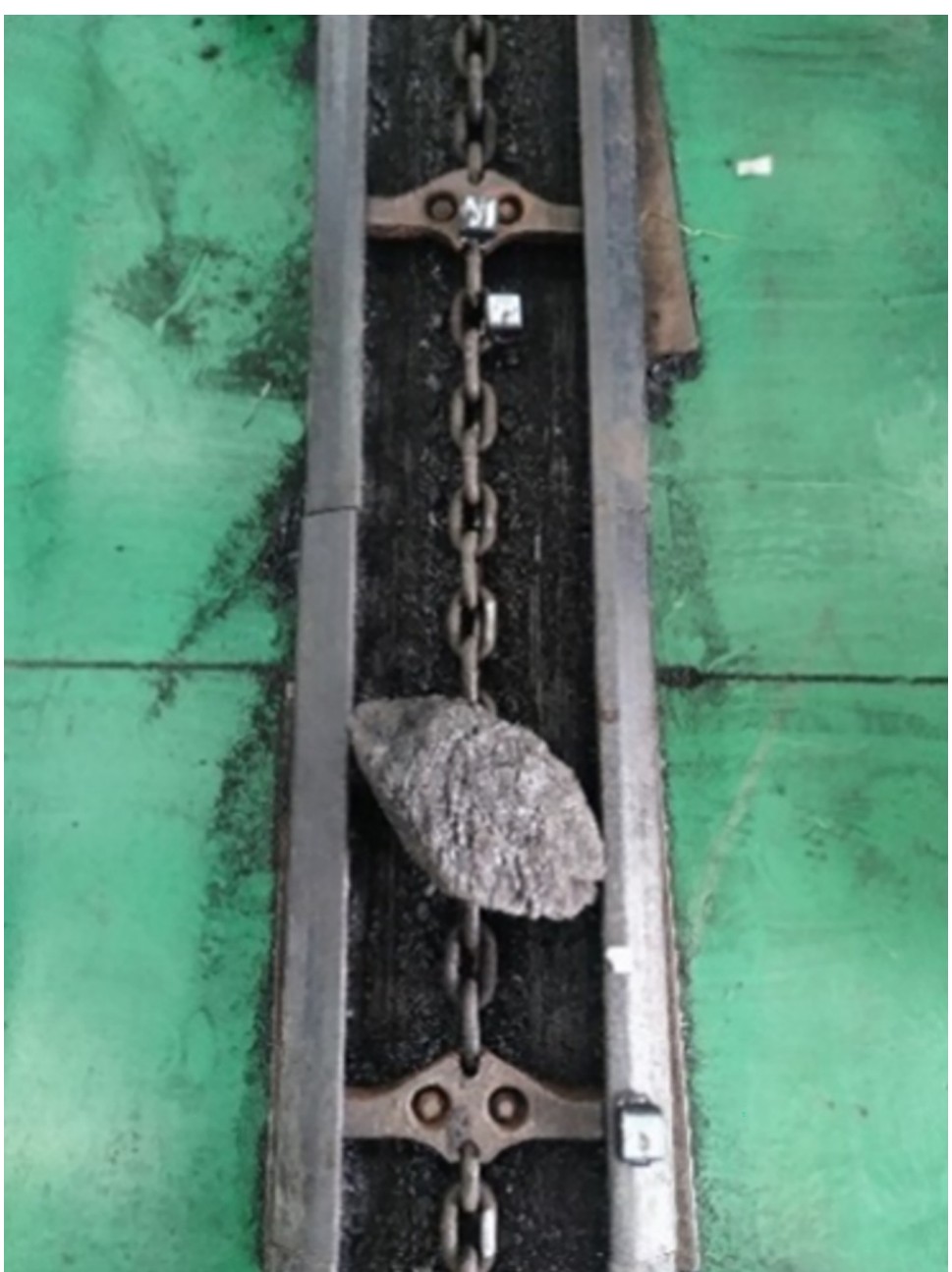

**Fig 4. Schematic diagram of coal block impact scraper conveyor.**

of scraper increased from 0.942m/s$^2$ to 2.625m/s$^2$, which increased 178.66%; the maximum acceleration of central groove increased from 0.065m/s$^2$ to 0.295m/s$^2$ with an increase of 353.85%. Secondly, in the Y direction, the maximum acceleration of chain ring increased from

**Table 5. The average value and standard deviation of acceleration in X, Y and Z directions.**

| | Average value (m/s$^2$) | | | Standard deviation (m/s$^2$) | | |
|---|---|---|---|---|---|---|
| Chain speed | Chain ring | Scraper | Central groove | Chain ring | Scraper | Central groove |
| 0.047 m/s | 2.363 | 2.748 | 0.495 | 1.789 | 0.962 | 0.408 |
| 0.259 m/s | 14.153 | 8.772 | 0.981 | 3.959 | 2.004 | 0.475 |

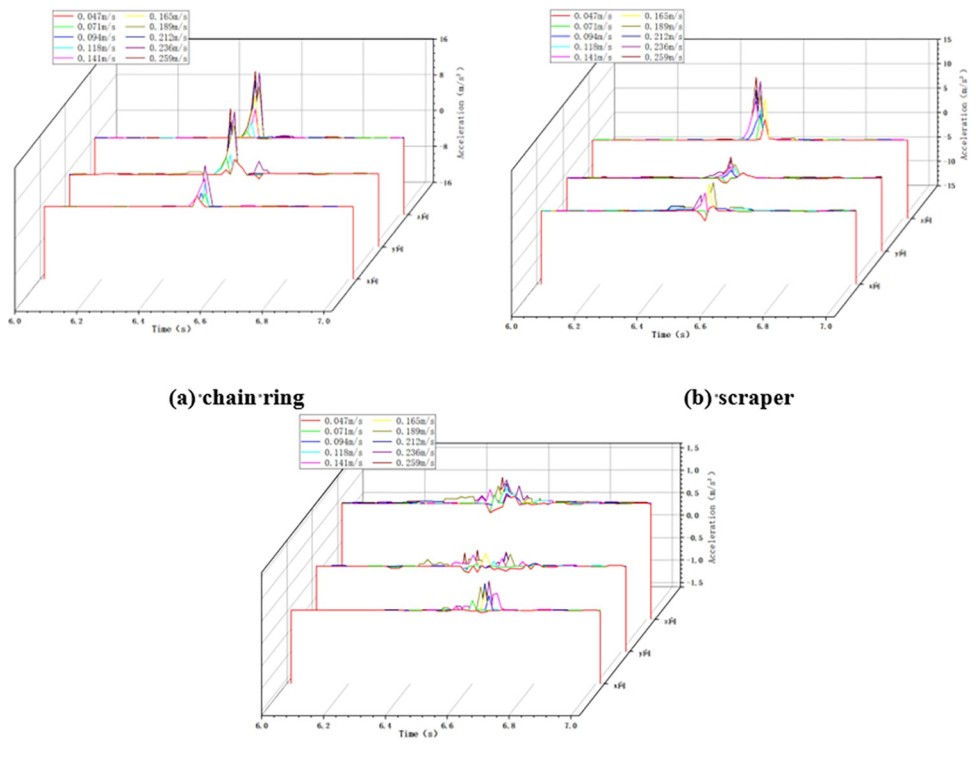

**(a) chain ring**

**(b) scraper**

**(c) central groove**

**Fig 5. Acceleration changes in variable chain speed conditions.**

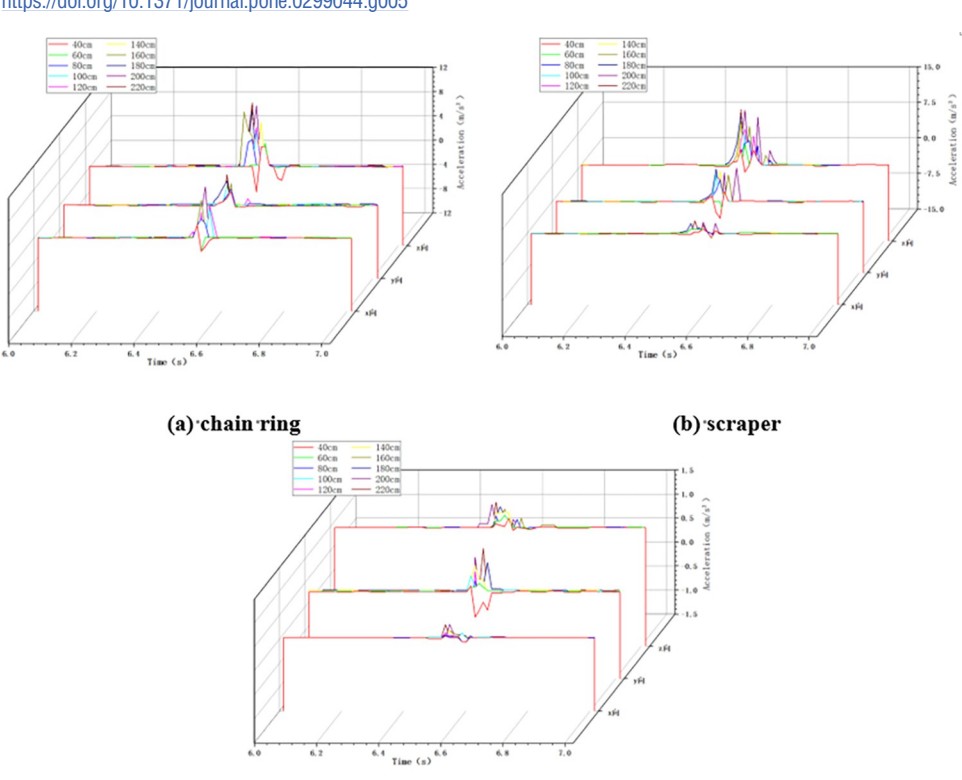

**(a) chain ring**

**(b) scraper**

**(c) central groove**

**Fig 6. Acceleration changes in variable height conditions.**

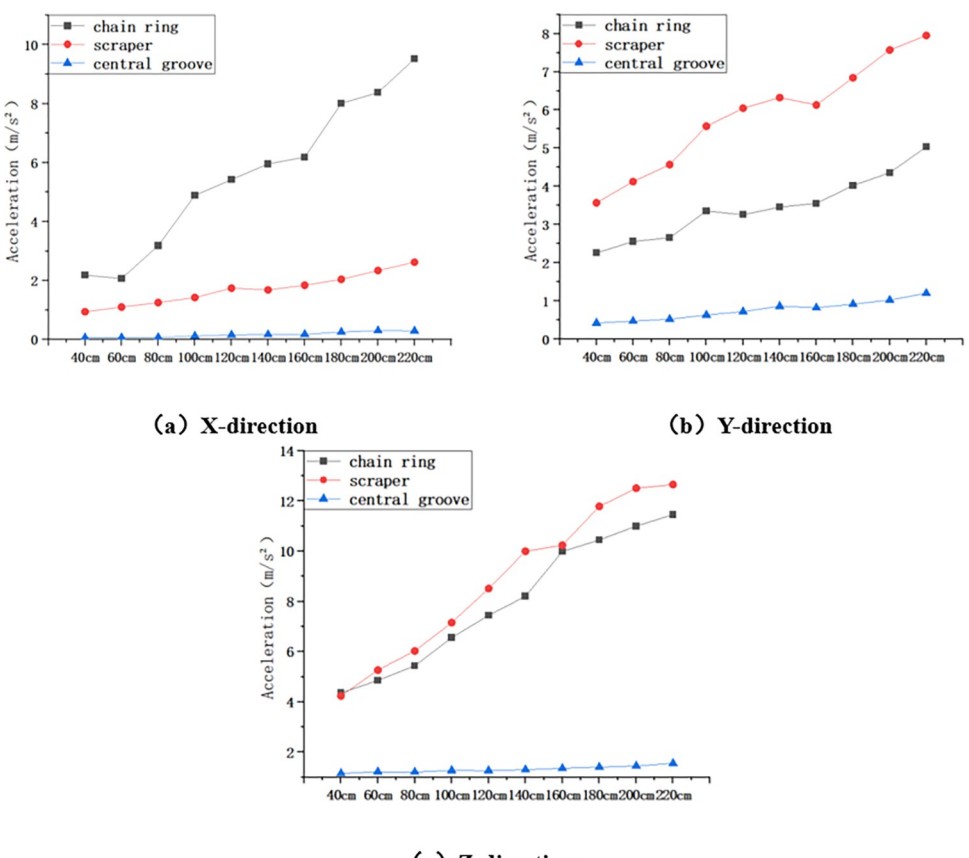

**Fig 7. Maximum acceleration in X, Y and Z directions for variable height conditions.**

2.258m/s$^2$ to 5.036m/s$^2$, which increased by 123.03%; the maximum acceleration of scraper increased from 3.572m/s$^2$ to 7.952m/s$^2$, which increased by 122.62%; and the maximum acceleration of the central groove increased from 0.425m/s$^2$ to 1.205m/s$^2$, which increased by 183.53%. Finally, in the Z direction, the maximum acceleration of the chain ring increased from 4.361m/s$^2$ to 11.458m/s$^2$, which is an increase of 162.74%; that of the scraper increased from 4.242m/s$^2$ to 12.655m/s$^2$, which is an increase of 198.33%; and that of the central groove increased from 1.159m/s$^2$ to 1.555m/s$^2$, which is an increase of 34.17%. Where the largest acceleration fluctuation in the X direction is for the chain ring, and the largest acceleration fluctuation in the Y and Z directions is for the scraper.

**4.1.3 Variable mass condition.** In the variable mass impact condition, the chain speed of the scraper conveyor is set to 0.118m/s, the impact height is set to 100cm, and the mass of the impact load is changed by changing the size of the coal block. The changes of acceleration in X, Y and Z directions at the three positions of the impacted chain ring, scraper and central groove of the scraper conveyor under the impact of different mass loads were collected as shown in Fig 8 below.

As shown in Fig 8 above, similar to the variable chain speed condition and the variable height condition, the three-direction acceleration of the chain drive system fluctuates drastically from zero when subjected to the impact of falling coal, and the greater the mass of the impact load, the greater the degree of fluctuation of the acceleration of the chain drive system.

The maximum acceleration of the chain drive system under the impact of different mass loads is statistically obtained, and the maximum acceleration in the X, Y, and Z

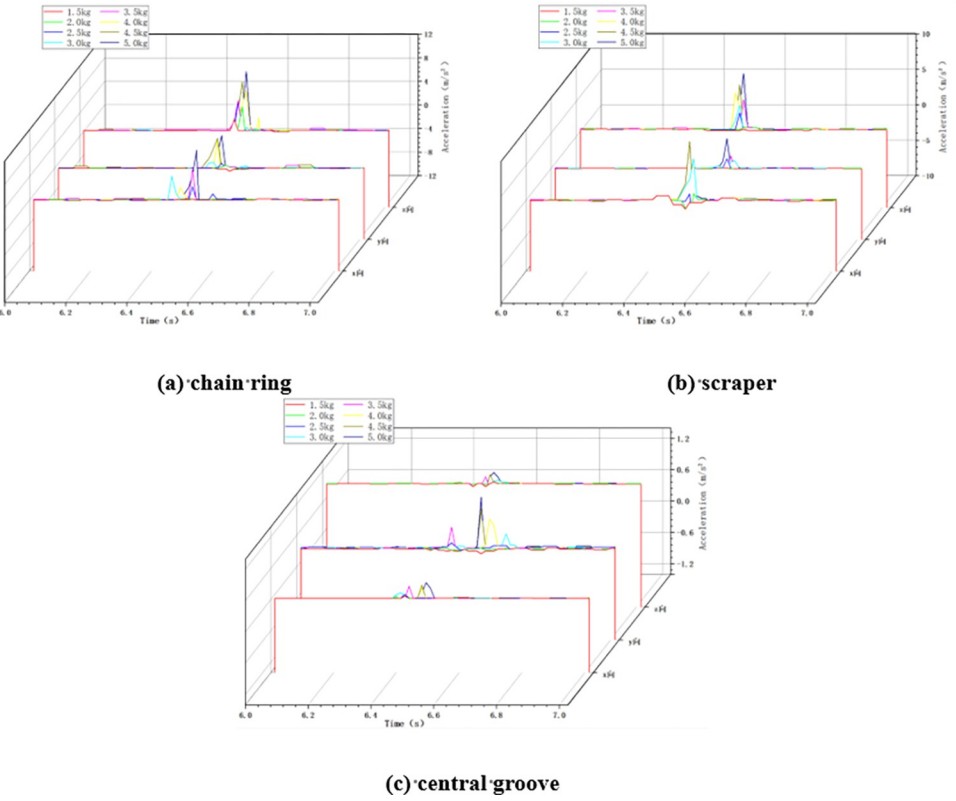

**Fig 8. Acceleration changes in variable mass conditions.**

directions during the increase of the mass of the impact load from 1.5 kg to 5 kg is shown in Fig 9 below.

As shown in Fig 9 above, firstly, in the X direction, the maximum acceleration of chain ring increases from $0.151m/s^2$ to $8.382m/s^2$, which is an increase of 5450.99%; the maximum acceleration of scraper increases from $1.257m/s^2$ to $8.128m/s^2$, which is an increase of 546.62%; and the maximum acceleration of central groove increases from $0.066m/s^2$ to $0.315m/s^2$, which is an increased by 377.27%. Secondly, in the Y direction, the maximum acceleration of chain ring increased from $0.458m/s^2$ to $5.512m/s^2$, which increased by 1103.49%; the maximum acceleration of scraper increased from $0.045m/s^2$ to $4.197m/s^2$, which increased by 9226.67%; and the maximum acceleration of central groove increased from $0.305m/s^2$ to $1.325m/s^2$, which increased by 334.43%. Finally, in the Z direction, the maximum acceleration of chain ring increased from $2.855m/s^2$ to $11.011m/s^2$, which is an increase of 285.67%; the maximum acceleration of scraper increased from $0.998m/s^2$ to $8.824m/s^2$, which is an increase of 784.17%; and the maximum acceleration of central groove increased from $0.758m/s^2$ to $1.169m/s^2$, which is an increase of 54.22%.

In summary, it can be seen that with the increase of the chain speed, impact height and mass of impact load of the scraper conveyor, the maximum acceleration in X, Y and Z directions of the chain ring, scraper and central groove in the time of impact show an increasing trend, which indicates that the chain speed, impact height and mass of impact load have a certain degree of excitation of the vibration in the three directions of the chain drive system.

In the above impact test, the maximum acceleration fluctuation of the chain drive system appeared in the Z-direction, and this phenomenon occurs because the direction of the coal block impacting the chain drive system is the vertical direction parallel to the Z-direction, so

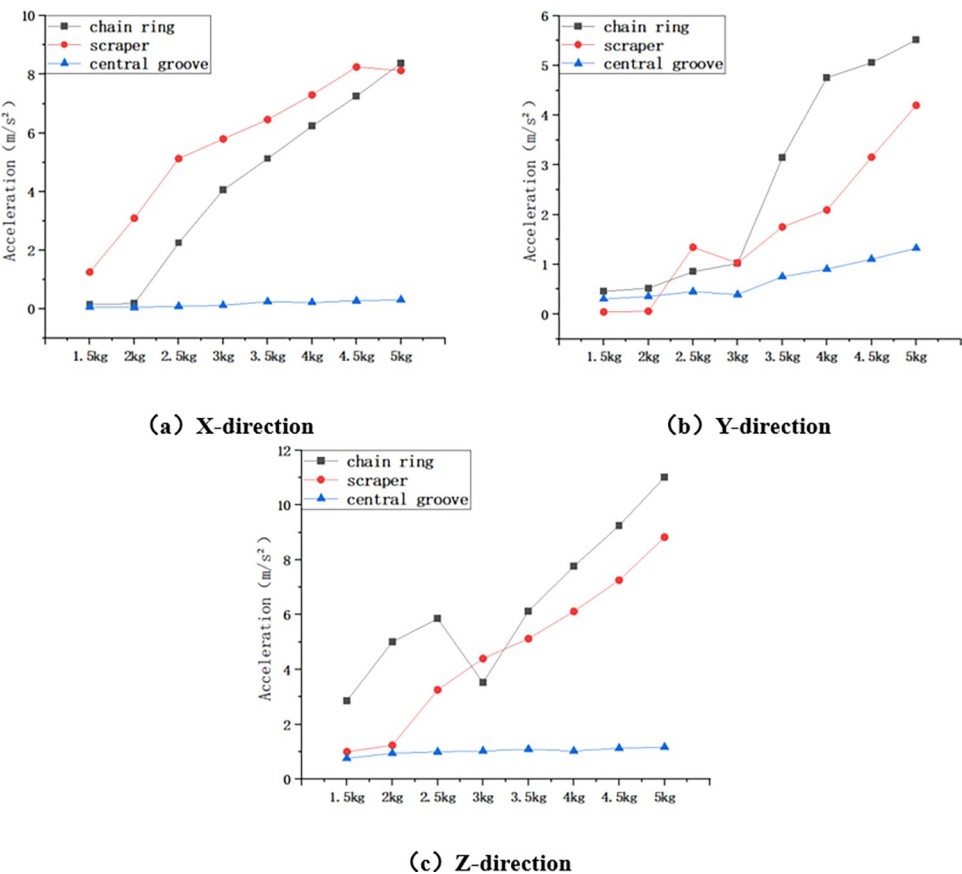

**Fig 9. Maximum acceleration in X, Y and Z directions for variable mass conditions.**

the Z-direction vibration of the chain ring is the most violent under the action of the vertical impact force. It is then known that longitudinal vibration is the main cause of chain ring failure when the scraper conveyor is subjected to impact under no-load condition.

## 4.2 Impact chain ring under loaded conditions

According to the different impact locations, the impact test under loaded conditions was divided into two parts: impacting the chain ring in front of the coal pile transported by the scraper conveyor, and directly impacting the coal pile transported by the scraper conveyor. The impact test was first carried out on the chain ring in front of the transported coal pile, in which the load between every two scraper sections on the scraper conveyor was fixedly set to a 2kg coal pile, and the loaded coal pile was loaded into the scraper conveyor by manual weighing followed by manual loading. The load schematic is shown in Fig 10 below, and the specific schematic of a coal block impacting a scraper conveyor is shown in Fig 11 below.

**4.2.1 Variable chain speed condition.** Similar to the no loaded condition, in the loaded variable chain speed impact condition, the impact height is taken as 100cm and the impact load mass is taken as 5kg. When the scraper conveyor is impacted under different chain speed conditions, the changes of acceleration in X, Y and Z directions at the three positions of the impacted chain ring, scraper and central groove are collected, as shown in Fig 12 below.

**4.2.2 Variable height condition.** In the variable height impact condition, the chain speed of the scraper conveyor is set to 0.118m/s and the mass of impact load is taken as 5kg. When

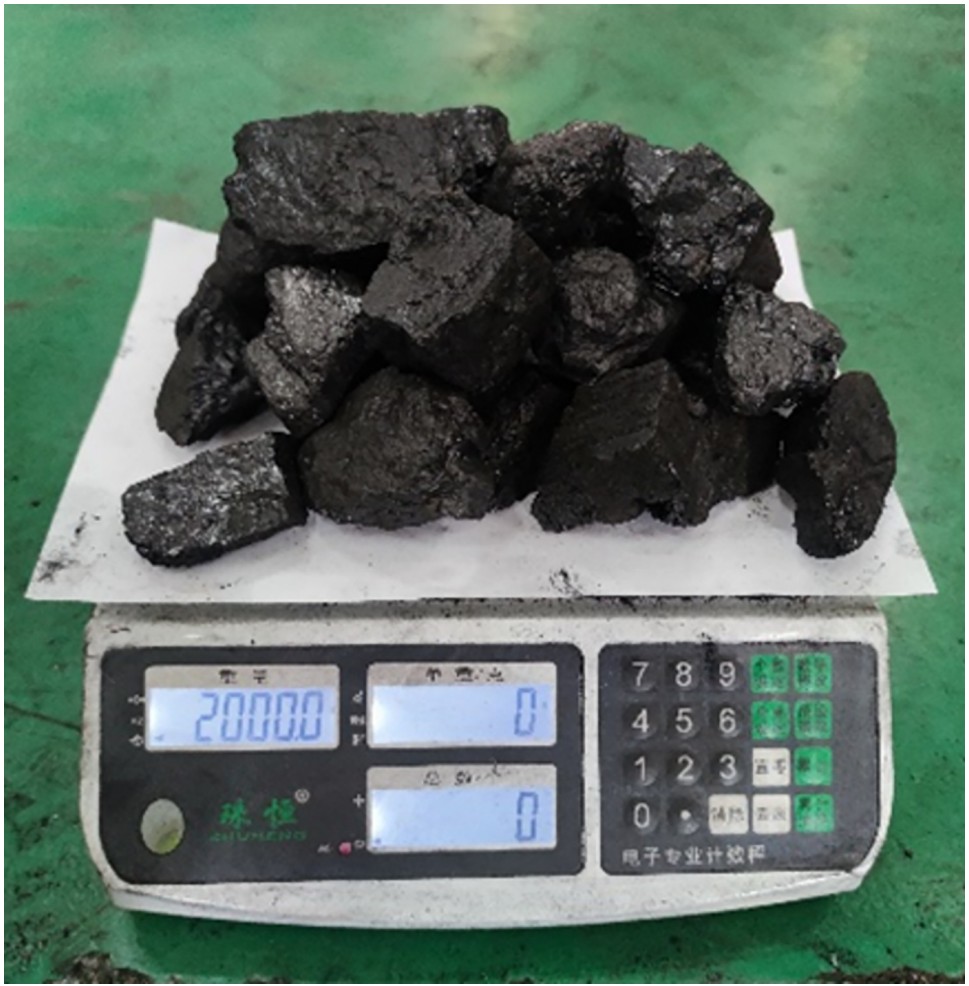

**Fig 10. Loaded coal pile diagram.**

the scraper conveyor is impacted at different impact heights, the X, Y and Z acceleration changes at the three positions of the impacted chain ring, scraper and central groove are collected, as shown in Fig 13 below.

**4.2.3 Variable mass condition.** In the loaded variable mass impact condition, the chain speed of the scraper conveyor was set to 0.118m/s and the impact height was set to 100cm. The changes of acceleration in X, Y and Z directions of the chain ring, scraper and central groove of the scraper conveyor under the impact of different masses of coal lumps were collected as shown in Fig 14 below.

It can be seen from Figs 12–14 above that, similar to the no-load condition, in the loaded impact chain ring condition, when the scraper conveyor is not impacted by the falling coal, the three-way acceleration of the chain drive system is basically zero, and the acceleration fluctuates drastically at the moment of the impact of the falling coal. And with the increase of chain speed, impact height and impact load mass, the three-direction acceleration at the three positions of the chain drive system of the scraper conveyor also shows an increasing trend, which indicates that the chain speed, impact height and impact load mass still have an excitation effect on the three-direction vibration of the chain drive system in the loaded working condition.

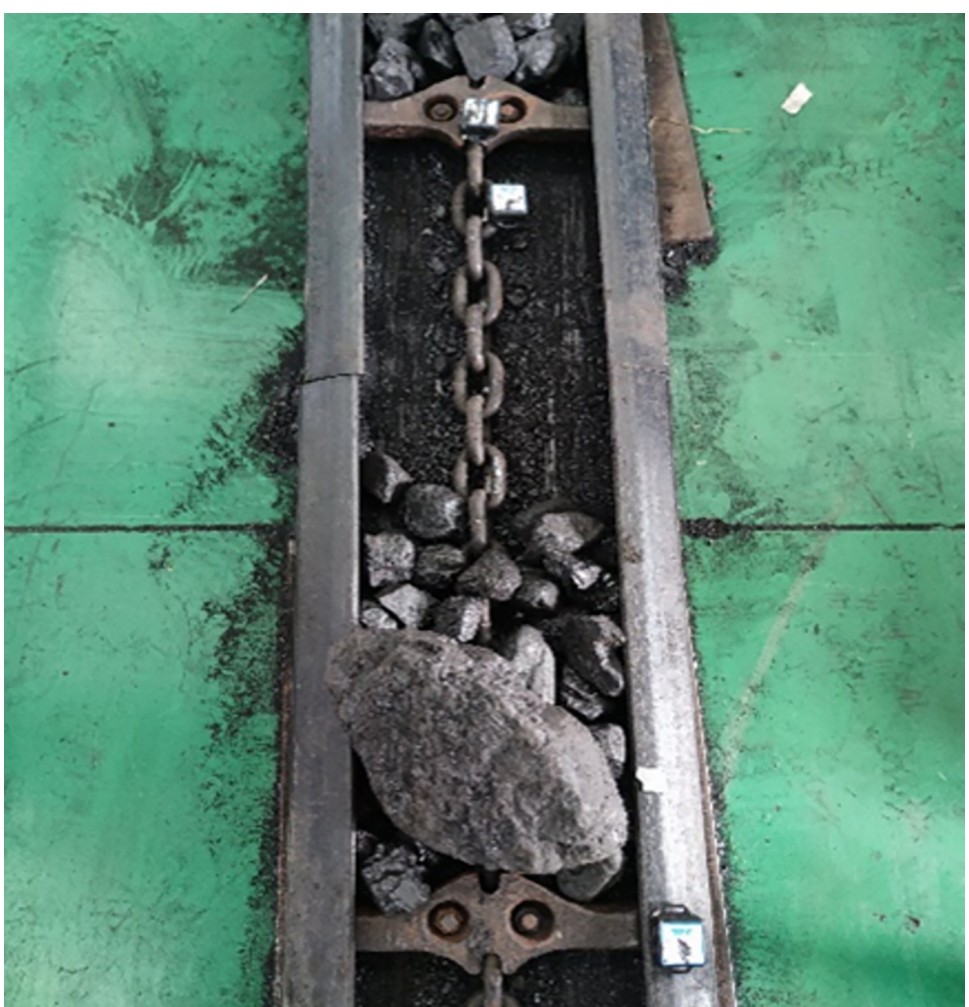

**Fig 11. Schematic diagram of coal block impact scraper conveyor.**

**4.2.4 Comparative analysis of no-load condition and loaded condition.** Taking the variable chain speed condition as an example, the average acceleration of chain ring, scraper and central groove in the process of increasing the chain speed from 0.047m/s to 0.259m/s is compared between the loaded impact chain ring condition and the no-load condition in the X, Y and Z directions, respectively, as shown in Fig 15 below.

Firstly, in the X direction, the average acceleration of chain ring changes from 6.437m/s$^2$ under no-load condition to 3.988m/s$^2$ under loaded condition, which is reduced by 38.05%; the scraper changes from 5.082m/s$^2$ under no-load condition to 3.019m/s$^2$ under loaded condition, which is reduced by 40.59%; the central groove changes from 0.451m/s$^2$ under no-load condition to 1.309m/s$^2$ under loaded condition, which is increased by 190.24%.

Secondly, in the Y-direction, the average acceleration of chain ring changes from 7.876m/s$^2$ under no-load condition to 6.901m/s$^2$ under loaded condition, which is reduced by 12.38%; the scraper changes from 2.418m/s$^2$ under no-load condition to 3.509m/s$^2$ under loaded condition, which is increased by 45.12%; and the central groove changes from 0.476m/s$^2$ under no-load condition to 0.533m/s$^2$ under loaded condition, which is increased by 11.97%.

Finally, in the Z direction, the average acceleration of chain ring changes from 8.547m/s$^2$ in no-load condition to 7.909m/s$^2$ in loaded condition, which is reduced by 7.46%; the scraper

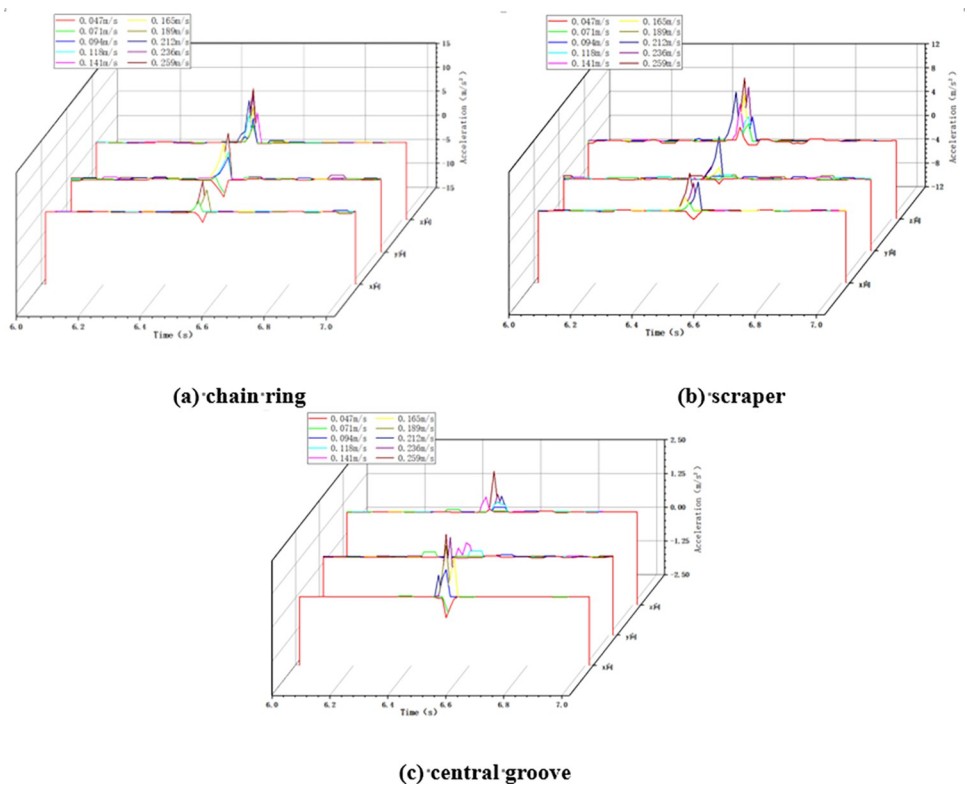

**Fig 12. Acceleration changes in variable chain speed conditions.**

changes from 9.128m/s$^2$ in no-load condition to 7.238m/s$^2$ in loaded condition, which is reduced by 20.71%; and the central groove changes from 1.326m/s$^2$ in no-load condition to 1.378m/s$^2$ under loaded condition, which is increased by 3.92%.

By comparing the average acceleration of the chain drive system subjected to impact under the loaded impact chain ring condition and the unloaded condition, it is found that in the three test conditions of variable chain speed, variable impact height, and variable impact load mass, the three-directional acceleration at the three positions under the loaded condition is

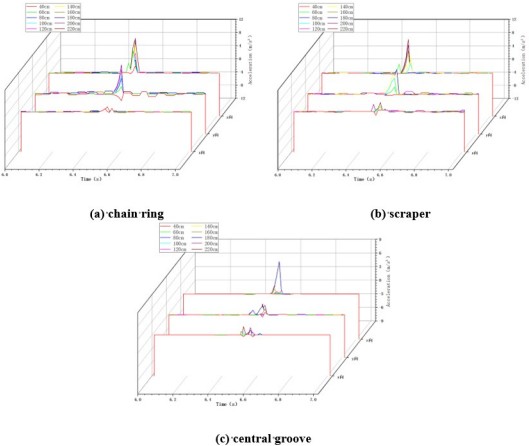

**Fig 13. Acceleration changes in variable height conditions.**

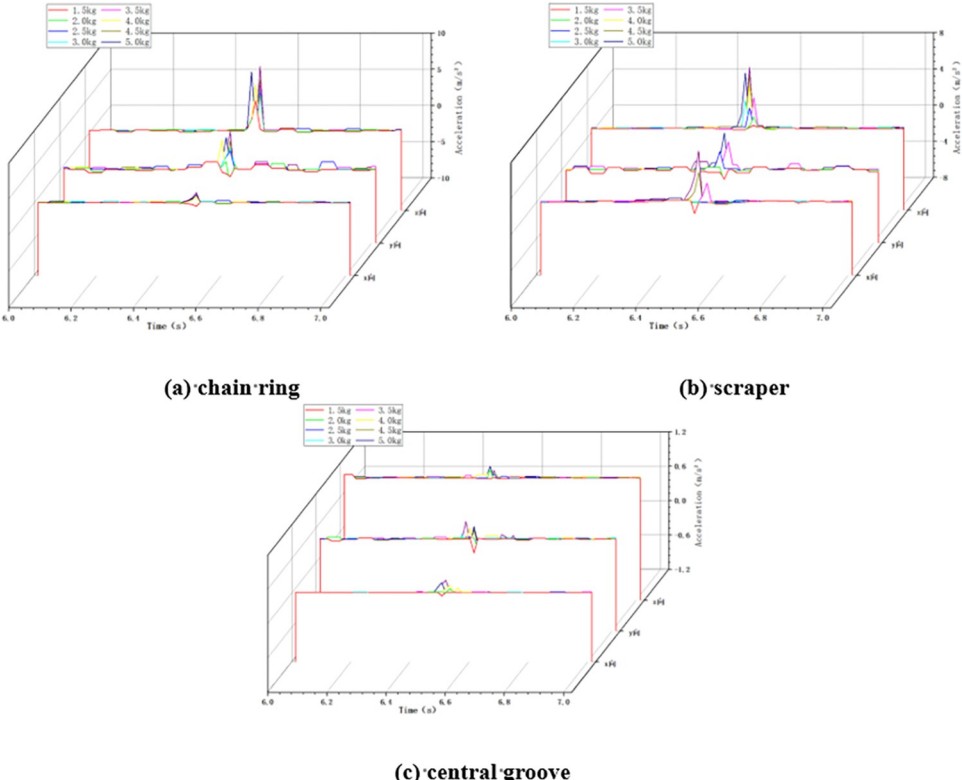

**Fig 14. Acceleration changes in variable mass conditions.**

relatively small compared with that under the unloaded condition. This is due to the fact that the coal pile transported on the scraper conveyor plays a certain stabilizing effect on the chain drive system and reduces the vibration of the chain drive system in all directions when it is subjected to impact, which indicates that the coal pile transported on the scraper conveyor has a certain inhibiting effect on the vibration of the chain drive system.

## 4.3 Impact coal pile under loaded conditions

Next is the second part of the loaded condition, where the impact test is performed directly on the loaded coal pile transported by the scraper conveyor, and a schematic diagram of the coal block impacting the scraper conveyor is shown in Fig 16 below.

 **4.3.1 Variable chain speed condition.**  In this variable chain speed condition, the impact height was set to 100cm and the mass of impacted coal block was set to 5kg. The X, Y and Z accelerations of the chain ring, scraper and central groove were collected when the scraper conveyor was impacted at different chain speeds, as shown in Fig 17 below.

 **4.3.2 Variable height condition.**  In the variable height impact condition, the chain speed of the scraper conveyor was set to 0.118 m/s and the impact load mass was set to 5 kg. The changes of acceleration in X, Y and Z directions of chain ring, scraper and central groove when the scraper conveyor is impacted under different impact height conditions are shown in Fig 18 below.

 **4.3.3 Variable mass condition.**  In the variable mass impact condition, the chain speed of the scraper conveyor was set to 0.118m/s and the impact height was fixed at 100cm. The variation of acceleration in X, Y and Z directions of chain ring, scraper and central groove when the scraper conveyor is impacted by loads of different masses is shown in Fig 19 below.

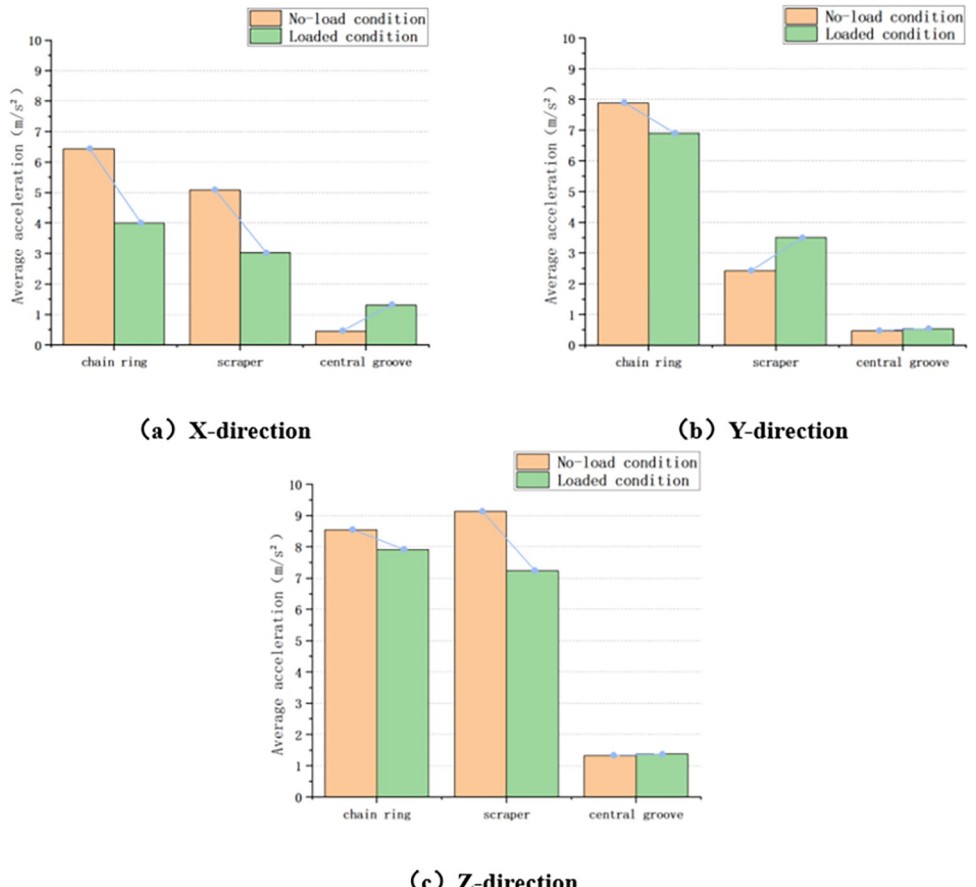

(a) X-direction (b) Y-direction

(c) Z-direction

**Fig 15. Comparison of loaded conditions and no-loaded conditions.**

As can be seen from the above Figs 17–19, with the increase of chain speed, impact height and impact load mass, the three-directional acceleration of the chain drive system likewise tends to increase, which further verifies the conclusion that the chain speed, impact height and impact load mass have an excitatory effect on three-directional vibration of the on-chain drive system.

**4.3.4 Comparison of impact chain ring and impact coal pile under loaded conditions.** Also take the variable chain speed condition as an example, compare the average acceleration of chain ring, scraper and central groove in the process of increasing the chain speed from 0.047m/s to 0.259m/s in the three directions of X, Y and Z, respectively, in the case of loaded impact chain ring and loaded impact coal pile, as shown in Fig 20 below.

Firstly, in the X direction, the average acceleration of chain ring changed from 3.988m/s$^2$ under the condition of impacting chain ring to 3.064m/s$^2$ under the condition of impacting coal pile, which decreased by 23.17%; the average acceleration of scraper changed from 3.019m/s$^2$ to 2.473m/s$^2$, which decreased by 18.09%; the average acceleration of central groove changed from 1.309m/s$^2$ to 0.565m /s$^2$, reduced by 56.8%.

Secondly, in the Y direction, the average acceleration of chain ring changed from 6.901m/s$^2$ under the condition of impacting chain ring to 6.044m/s$^2$ under the condition of impacting coal pile, which decreased by 12.42%; the average acceleration of scraper changed from 3.509m/s$^2$ to 4.326m/s$^2$, which increased by 23.28%; and the average acceleration of central groove changed from 0.533m/s$^2$ to 0.475m /s$^2$, an increase of 10.88%.

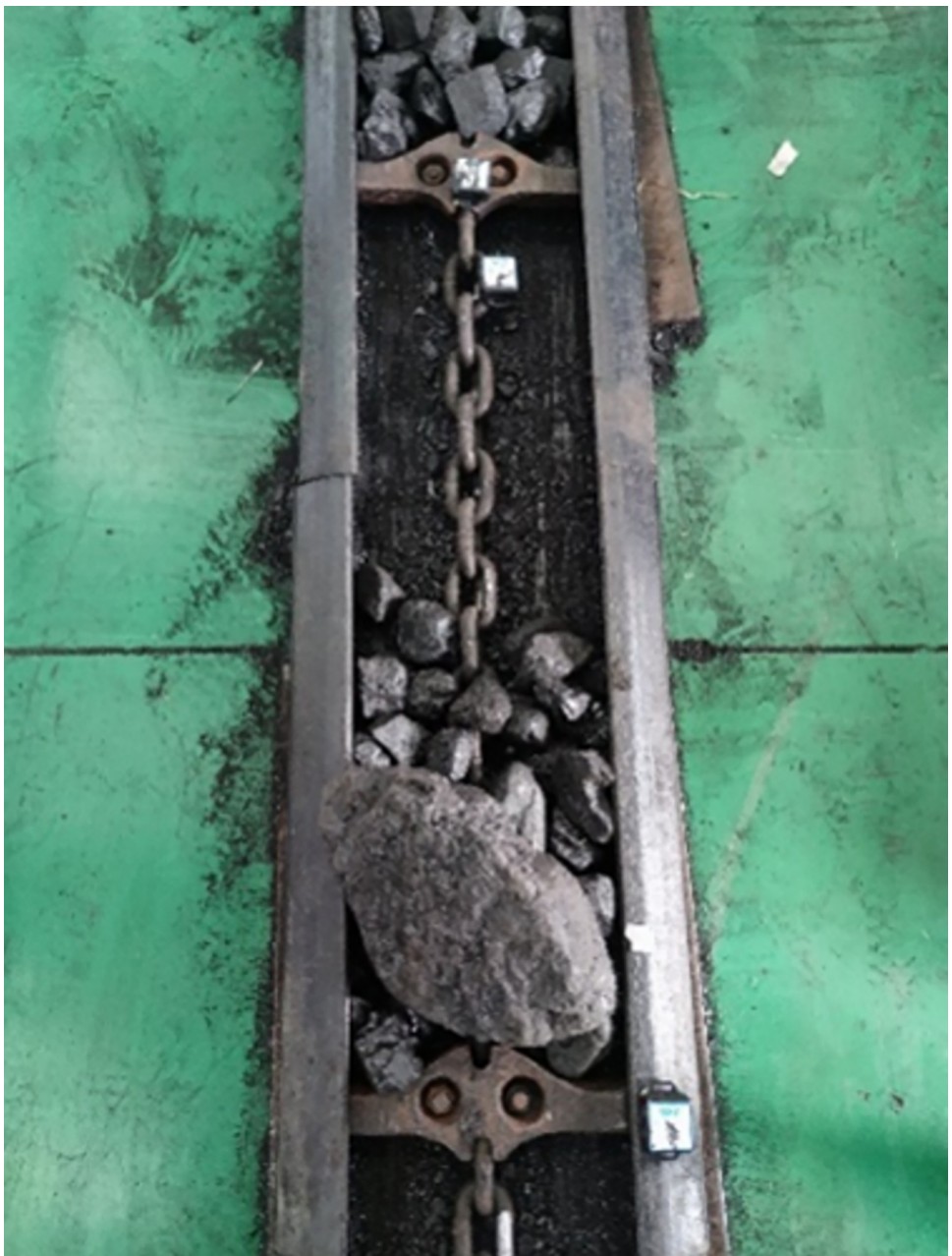

**Fig 16. Schematic diagram of coal block impact scraper conveyor.**

Finally, in the Z direction, the average acceleration of chain ring changed from 7.909m/s$^2$ under the condition of impacting chain ring to 7.714m/s$^2$ under the condition of impacting coal pile, which decreased by 2.47%; the average acceleration of scraper changed from 7.238m/s$^2$ to 5.983m/s$^2$, which decreased by 17.34%; and the average acceleration of central groove changed from 1.378m/s$^2$ to 1.263m/s$^2$, which decreased by 8.35%. s$^2$, a decrease of 8.35%.

Through the above comparison, it is found that the three-directional acceleration at each position when impacting the loaded coal pile is further reduced compared with that when impacting the chain ring condition, except that there is a small increase in the average

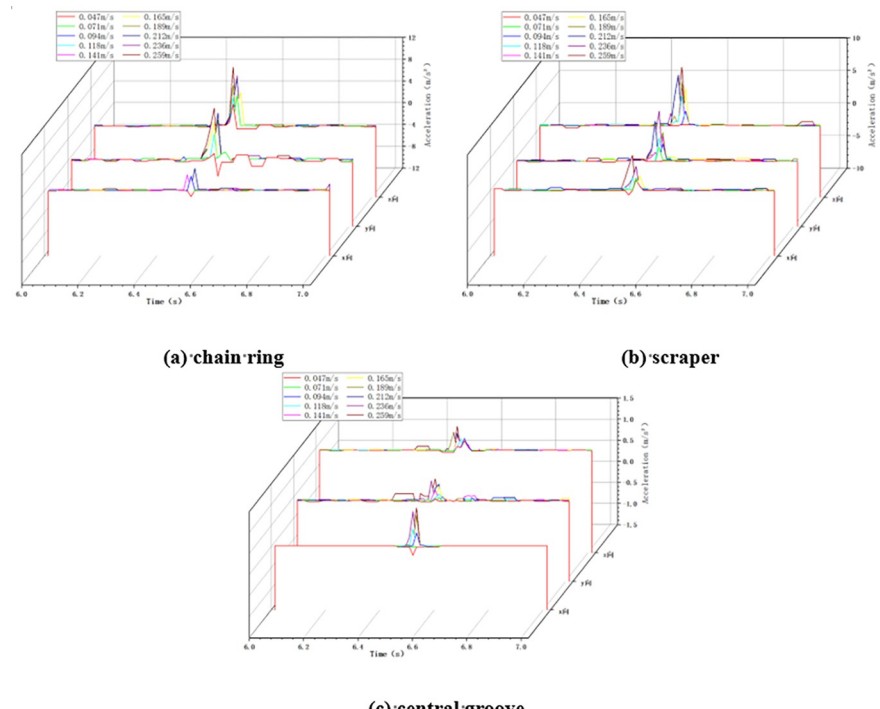

**(a) chain ring**

**(b) scraper**

**(c) central groove**

**Fig 17. Acceleration changes in variable chain speed conditions.**

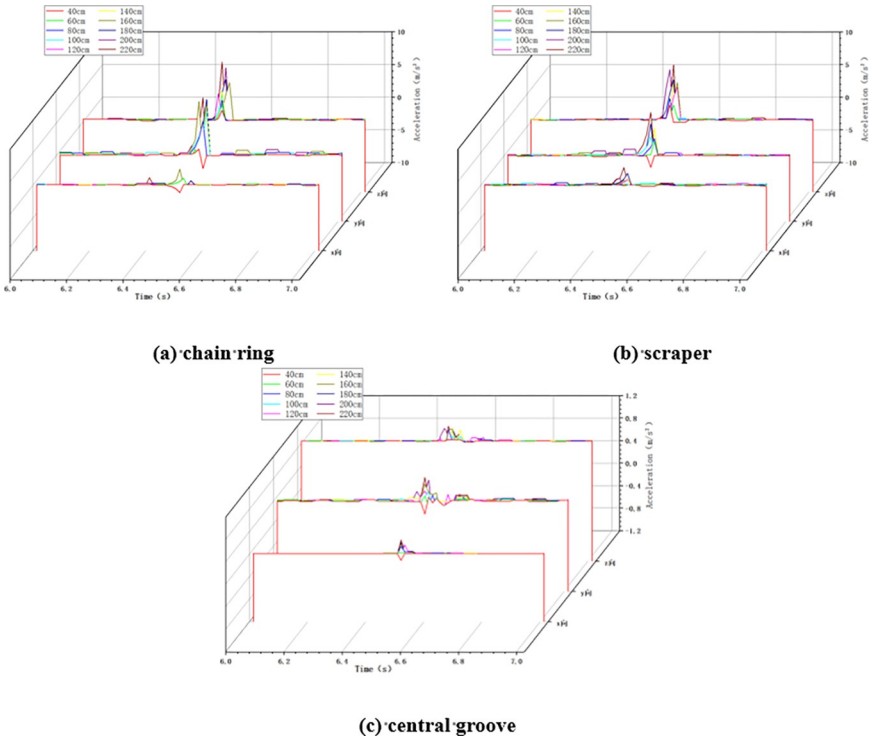

**(a) chain ring**

**(b) scraper**

**(c) central groove**

**Fig 18. Acceleration changes in variable height conditions.**

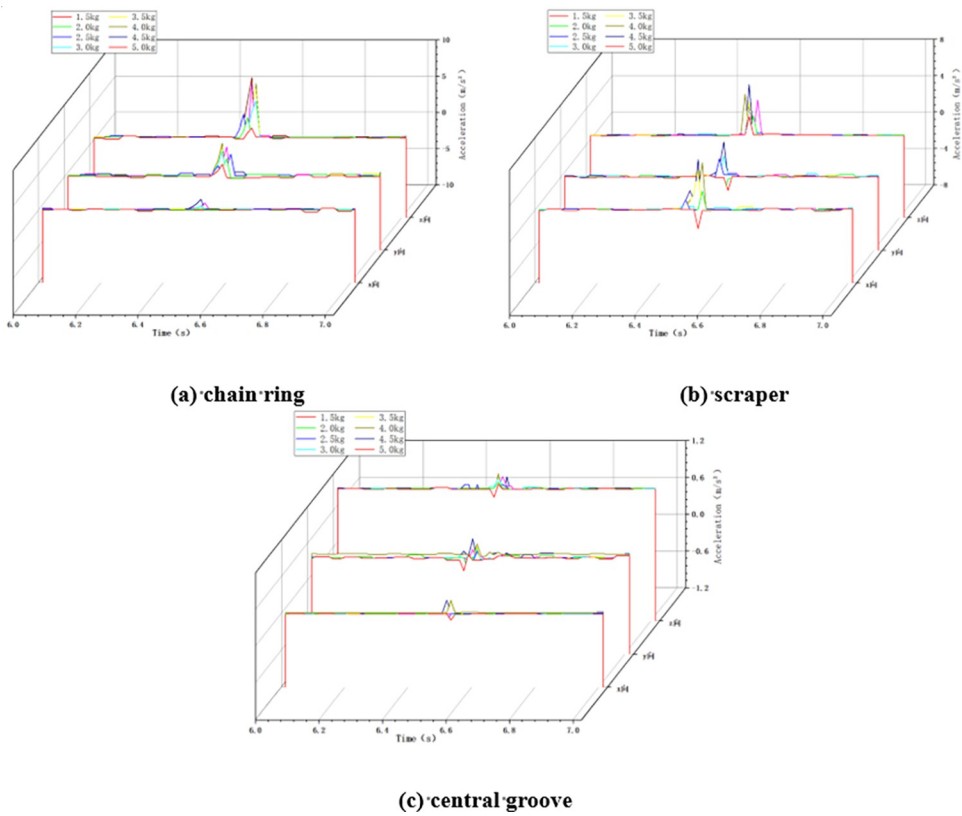

**Fig 19. Acceleration changes in variable mass conditions.**

acceleration of the scraper and the central groove in the Y-direction. This is because when the scraper conveyor is impacted by the coal, the coal pile on the scraper conveyor plays a certain buffer effect on the impact, reducing the impact of the coal for the scraper conveyor, and the transportation of the coal pile of the vibration of the chain also plays a certain role in suppressing. Therefore, the acceleration fluctuation in the case of impacting coal pile is smaller compared with that in the case of impacting chain ring, which also further verifies the conclusion that the coal pile transported on the scraper conveyor has a certain suppression effect on the vibration of the chain drive system.

## 5 Conclusions

In this paper, the influence of different chain speeds, different impact heights and different impact load masses on the dynamic characteristics of the chain drive system of the scraper conveyor under the impact condition is investigated by constructing the impact test bench of the scraper conveyor, and the following conclusions are obtained by collecting and analyzing the test data:

1. When the scraper conveyor is impacted by the falling coal, the chain speed, impact height and impact load mass all have an excitation effect on the vibration of the chain drive system, in which the chain speed has the most obvious effect, followed by the impact height and finally the mass of the impact load.

2. When the scraper conveyor is impacted by falling coal, the chain drive system vibrates most violently in the Z direction. Therefore, the following conclusion can be deduced: the

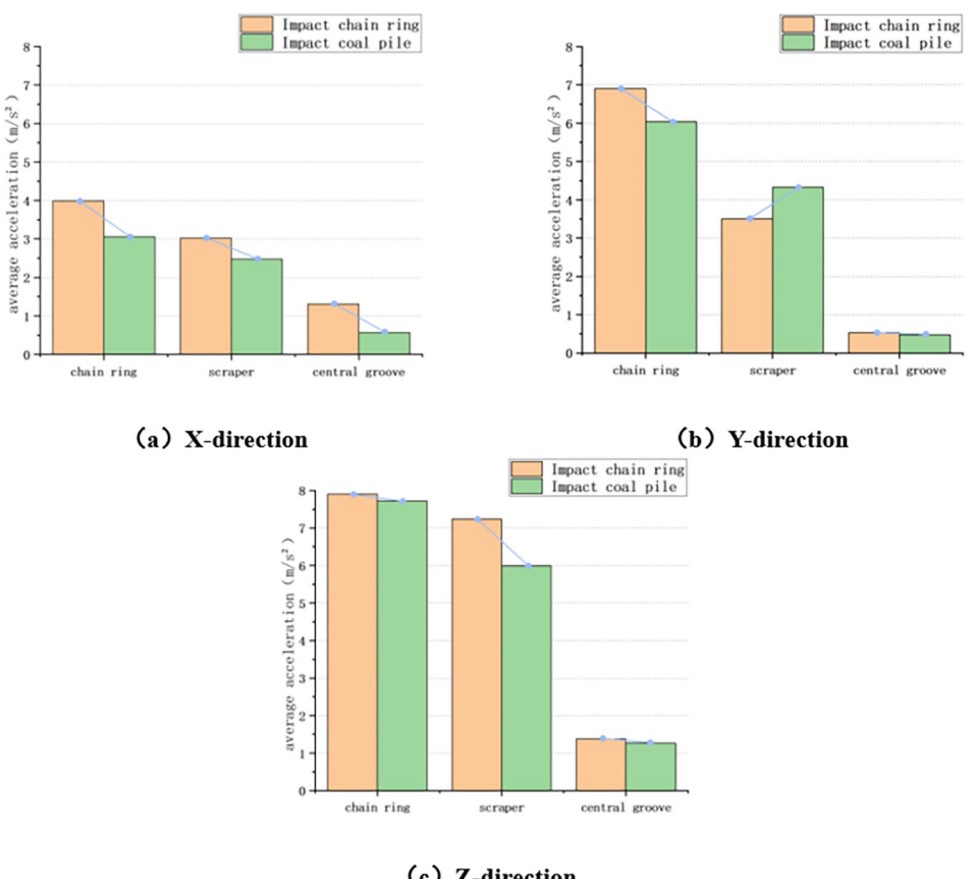

**Fig 20. Comparison of impact chain ring and impact coal pile under loaded conditions.**

longitudinal vibration of the scraper conveyor is the main reason for the failure of the chain drive system under impact conditions.

3.  When the scraper conveyor is impacted by the falling coal, the vibration of the chain ring is the most violent, followed by the scraper, and finally the central groove. Therefore, under the impact condition of scraper conveyor, the chain ring and chain are the first part to fail.

4.  Compared with the unloaded condition, the three-directional acceleration fluctuations of the chain drive system are reduced when the scraper conveyor is impacted by the falling coal under the loaded condition, which indicates that the transported coal pile has a certain suppression effect on the vibration of the chain drive system.

This paper investigates the vibration of scraper conveyor when it is impacted under multiple working conditions, which is of great significance to the stable operation and structural optimization of scraper conveyor. However, the present study is limited by the conditions of the test bench, the values of mass and height of impact load are conservative, and the effect of terrain change on the dynamics of the scraper conveyor under impact conditions is not studied. Future research will further improve the impact test bench of scraper conveyor by using a larger mass of impact load and a higher impact height for impact test, it can more realistically simulate the real working conditions and improve the reliability and accuracy of the study. At the same time, it is necessary to add the test conditions of terrain change, to study the influence of factors such as surface undulation on the dynamic characteristics of the scraper conveyor

under impact conditions, to further improve the research on the dynamic characteristics of the scraper conveyor under impact conditions.

## Supporting information

**S1 File.**
(ZIP)

**S2 File.**
(ZIP)

**S3 File.**
(ZIP)

**S4 File.**
(ZIP)

## Author Contributions

**Conceptualization:** Qingliang Zeng, Yuqi Zhang.

**Data curation:** Qingliang Zeng, Jiexu Cui.

**Funding acquisition:** Shoubo Jiang.

**Investigation:** Shoubo Jiang.

**Methodology:** Jinwang Lv.

**Project administration:** Jinwang Lv.

**Resources:** Qiang Zhang.

**Supervision:** Qiang Zhang.

**Validation:** Yuqi Zhang, Wei Qu.

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
