## [Decision Letter · Decision Letter 0]

20 Sep 2023

PONE-D-23-28370Dynamic characteristics of scraper conveyor chain drive system under the impact condition of lump coalPLOS ONE

Dear Dr. Zeng,

Thank you for submitting your manuscript to PLOS ONE. After careful consideration, we feel that it has merit but does not fully meet PLOS ONE’s publication criteria as it currently stands. Therefore, we invite you to submit a revised version of the manuscript that addresses the points raised during the review process.

We look forward to receiving your revised manuscript.

Kind regards,

Brij Bhooshan Gupta

Academic Editor

PLOS ONE

Journal Requirements:

1. Please ensure that your manuscript meets PLOS ONE's style requirements, including those for file naming. The PLOS ONE style templates can be found at https://journals.plos.org/plosone/s/fileid=wjVg/PLOSOne_formatting_sample_main_body.pdf and https://journals.plos.org/plosone/s/file?id=ba62/PLOSOne_formatting_sample_title_authors_affiliations.pdf

Reviewers' comments:

Reviewer's Responses to Questions

**Comments to the Author**

1. Is the manuscript technically sound, and do the data support the conclusions?

Reviewer #1: Yes

Reviewer #2: Yes

2. Has the statistical analysis been performed appropriately and rigorously? 

Reviewer #1: Yes

Reviewer #2: Yes

3. Have the authors made all data underlying the findings in their manuscript fully available?

Reviewer #1: Yes

Reviewer #2: Yes

4. Is the manuscript presented in an intelligible fashion and written in standard English?

Reviewer #1: Yes

Reviewer #2: Yes

5. Review Comments to the Author

Reviewer #1: In this paper, the dynamic characteristics of scraper conveyor chain drive system when impacted by falling coal are investigated by means of test.

- Research Motivation: What motivated you to investigate the dynamic characteristics of scraper conveyor chain drive systems, specifically under the impact condition of lump coal?

-Significance of Scraper Conveyors: Could you explain why scraper conveyors are considered the most important transportation equipment in comprehensive mining, and how their efficiency impacts coal transportation?

- Research Methodology: How did you set up the impact test bench for scraper conveyor testing, and what parameters were considered, such as chain speed, impact height, and impact load mass?

- Dynamic Characteristics: What are the key dynamic characteristics you observed in the chain drive system when impacted by falling coal, and how do these characteristics affect the overall performance of the conveyor?

-Effect of Parameters: Can you elaborate on the effects of chain speed, impact height, and impact load mass on the dynamic characteristics of the chain drive system, both under no-load and loaded conditions?

- Vibration Analysis: What insights did you gain from the analysis of vibration within the chain drive system, including the chain ring, scraper, and central groove?

- Role of Loaded Coal: You mentioned that the loaded coal pile on the scraper conveyor acts as a suppressor of vibration. Could you explain how this suppression occurs and its implications for conveyor operation?

- Practical Significance: How can the findings of your study be practically applied to enhance the stable operation and structure optimization of scraper conveyors in mining operations?

- Future Research: Are there any specific areas of future research that you believe should be explored based on the results of your study or any limitations you encountered?

- Industry Impact: How might the mining industry benefit from a better understanding of the dynamic characteristics of scraper conveyor chain drive systems, particularly in terms of safety and efficiency?

- Research Contribution: What do you consider the main contribution of your research to the field of mining equipment and conveyor systems?

Reviewer #2: This research study investigated the dynamic characteristics of the scraper conveyor chain drive system when impacted by falling coal. In general, the paper is well written. There are some key comments to be considered by the authors before the recommendation of acceptance. Please refer to my comments as follows.

Comment 1. Revisit the paper to correct some typos and minor grammatical mistakes.

Comment 2. Abstract: Highlight the key research results and implications.

Comment 3. Ensure proper spacing between wordings and symbols throughout the paper. For example, correct “development[1].”, “equipment[2].”…

Comment 4. Section 1 Introduction:

(a) Update the list of references to cover more recent 2-year published articles. Elaborate on the results and limitations of existing works.

(b) Summarize the research contributions of the paper.

Comment 5. Section 2 Theoretical basis:

(a) Clarify if the formulation follows existing works as background information. If yes, include references.

(b) Ensure the presented content helps readers understand the rest of the technical details in the coming sections.

Comment 6. Section 3 Research Program:

(a) Add an introductory paragraph before Subsection 3.1.

(b) Justify if the models of scraper conveyors and sensors are common in the industry.

(c) The test schemes only covered limited settings. Justify if they align with the ranges in reality.

Comment 7. Section 4 Test results and analysis:

(a) Add an introductory paragraph before Subsection 4.1.

(b) Enhance the resolutions of all figures. Ensure no content is blurred.

(c) If there are subfigures, ensure the captions are included for all subfigures.

(d) Table 5, apart from average values, includes more information.

(e) Comparison between proposed work and existing works is expected.

Comment 8. Elaborate on future research directions.

6. PLOS authors have the option to publish the peer review history of their article (what does this mean?). If published, this will include your full peer review and any attached files.

Reviewer #1: No

Reviewer #2: **Yes: **Kwok Tai Chui

---

## [Author Response · Author response to Decision Letter 0]

9 Nov 2023

Dear Reviewers:

Thank you for reviewing our manuscript and for your valuable comments and suggestions. We are very grateful for your recognition of our research and we are more than willing to revise and improve the manuscript based on your suggestions.

Based on your opinions and suggestions, we hereby provide the following detailed response:

To Reviewer 1:

In this paper, the dynamic characteristics of scraper conveyor chain drive system when impacted by falling coal are investigated by means of test.

- Research Motivation: What motivated you to investigate the dynamic characteristics of scraper conveyor chain drive systems, specifically under the impact condition of lump coal?

First of all, the scraper conveyor is the most important transportation equipment in the comprehensive mining equipment, and it also is the track of the coal mining machine and the fulcrum of the hydraulic support, so to ensure the normal operation of the scraper conveyor is crucial to the coal mining work, and the chain drive system is the core sub-system of the scraper conveyor, so it’s necessary to carry out the study of the dynamic characteristics of the chain drive system of the scraper conveyor.

At the same time because of the coal mine environment is relatively harsh, so the working environment of the scraper conveyor is complex, there are often large scale coal rock collapsed to the scraper conveyor, which seriously affects the dynamics characteristics of the scraper conveyor, and in serious cases, it will even make the scraper conveyor broken chain, chain jamming and other failures. Based on this realistic background, I carried out a study on the dynamic characteristics of scraper conveyor under the impact condition of lump coal, and I hope that this study can provide help for the structural improvement and optimization of scraper conveyor.

-Significance of Scraper Conveyors: Could you explain why scraper conveyors are considered the most important transportation equipment in comprehensive mining, and how their efficiency impacts coal transportation?

The strength and rigidity of the body of the scraper conveyor is good, and it can transport the coal from the working environment with poor conditions to the place with better transportation conditions. Compared with the traditional transportation by manpower or vehicles, the scraper conveyor greatly improves the efficiency of coal mining, and at the same time, it also reduces the number of personnel on the working face, and guarantees the safety of the working face. In addition to its own coal transportation function, the scraper conveyor is also the track of the coal mining machine and the fulcrum of the hydraulic support, so the scraper conveyor is considered to be the most important transportation equipment in the comprehensive mining .

The efficiency of the scraper conveyor is very important for coal transportation. If the efficiency of the scraper conveyor is low, it will lead to the accumulation of coal on the coal mining face, which will affect the continuity of coal mining. In addition, the efficiency of the scraper conveyor also affects the efficiency of the whole coal production line, if the efficiency of scraper conveyor is low, it will cause the efficiency of the whole production line to drop, which will reduce the overall efficiency of coal production.

- Research Methodology: How did you set up the impact test bench for scraper conveyor testing, and what parameters were considered, such as chain speed, impact height, and impact load mass?

Based on the SGD320/17B scraper conveyor, this study designed the final impact test bench of the scraper conveyor by independently designing the large scale coal impact device, matching the sensor and data acquisition and analysis system.

In order to simulate the actual working environment of the scraper conveyor as much as possible, this study comprehensively considered various variable conditions that may be encountered during the operation of the scraper conveyor when setting up the impact test bench of the scraper conveyor, such as the height of the impact of the lump coal, the mass of the lump coal, the chain speed of the scraper conveyor, and the location of the impact of the lump coal. Based on this impact test bench of scraper conveyor, the height of lump coal impact can be adjusted by the hand winch and wire rope on the large scale coal impact device, the mass of lump coal can be adjusted by selecting different sizes of lumps, the chain speed of the scraper conveyor can be controlled by the frequency inverter, and the position of the lump coal impact can be controlled by changing the relative position of the lump coal impact device and the scraper conveyor. Therefore, this impact test bench of scraper conveyor can be used to simulate various actual working conditions.

- Dynamic Characteristics: What are the key dynamic characteristics you observed in the chain drive system when impacted by falling coal, and how do these characteristics affect the overall performance of the conveyor?

First of all, the impact of falling coal will lead to a sudden change in the load of the chain drive system. When a large scale coal falls and impacts the chain drive system, it creates an impact force on the drive chain, which causes the chain drive system to be subjected to a greater load, and this sudden change in load can adversely affect the life and reliability of the chain ring, scraper, central groove, and other drive system components.

Secondly, the impact of falling coal will cause violent vibration of the chain drive system. The impact of large scale coal on the scraper conveyor will generate vibration energy and transfer it to the whole transmission system through the chain, and this vibration will not only increase the noise and vibration level of the transmission system, but also generate additional load on the bearings, sprockets and other parts, which will reduce the working efficiency and life of the system.

Finally, the impact of falling coal may also lead to chain disengagement or chain breakage. The huge impact will make the chain jump or break in the transmission process, which will lead to the chain out of the sprocket groove, resulting in transmission system interruption and shutdown, which will adversely affect the normal operation and productivity of the conveyor.

To sum up, the chain drive system will face critical dynamic characteristics such as sudden load change, vibration increase and chain breakage when impacted by falling coal, which will negatively affect the performance of the conveyor and reduce the reliability, life and working efficiency of the system.

-Effect of Parameters: Can you elaborate on the effects of chain speed, impact height, and impact load mass on the dynamic characteristics of the chain drive system, both under no-load and loaded conditions?

When the chain drive system is impacted by falling coal under no-load condition, the chain speed, impact height and impact load mass will have different effects on the dynamic characteristics as follows:

1. Chain speed: when the chain speed is low, the impact force of falling coal is relatively small, while when the chain speed increases, the impact force of falling coal will become larger, and the higher chain speed will make the impact force more significant and produce greater impact and vibration on the chain and chain drive system components.

2. Impact height: Higher impact height means that the kinetic energy of lump coal is larger, and at the same time the impact chain will bring more impact force, so under the same conditions, higher impact height will increase the impact load of the chain drive system.

3. Impact load mass: A larger impact load mass will result in a greater impact force acting on the chain drive system, which will increase the stress and vibration of the chain and may lead to problems such as chain breakage or chain breakage.

When the chain drive system is impacted by falling coal under loaded condition, the influence of various factors on the dynamic characteristics is similar to that of no-load condition, and the various vibrations and impacts generated by falling coal under loaded condition are smaller than that of no-load condition.

- Vibration Analysis: What insights did you gain from the analysis of vibration within the chain drive system, including the chain ring, scraper, and central groove?

After analyzing the vibrations within the chain drive system, I came up with some of the following insights:

1. Vibration control: Controlling vibration in chain drive systems can improve system reliability and performance. For the vibration of chain and other components, vibration damping measures can be taken, such as the use of vibration-damping shims and the addition of vibration dampers. In addition, proper lubrication and maintenance can also reduce the noise generated by vibration and friction and improve transmission efficiency.

2. Material and design optimization: Optimizing the material and design of chain rings, scrapers and central grooves can reduce vibration problems, and choosing high-strength and low-wear materials can reduce chain vibration and loss. In addition, reasonable chain size and geometry design, including suitable size clearance and the contact area between chain and plate, can also improve the dynamic characteristics of chain drive system.

3. Regular inspection and maintenance: Regular inspection of the state and working conditions of the chain drive system is the key to preventing vibration problems. Checking the chain tension, degree of wear and looseness, and making timely adjustments and replacements the chain, as well as keeping the chain clean and lubricating it in time are important means of preventing vibration.

4. Optimization of operating conditions: Reasonable setting of the operating conditions of the chain drive system can also improve its vibration characteristics. For example, operational optimization such as controlling chain speed, avoiding overloads and sudden load changes will help reduce the occurrence of vibration problems.

- Role of Loaded Coal: You mentioned that the loaded coal pile on the scraper conveyor acts as a suppressor of vibration. Could you explain how this suppression occurs and its implications for conveyor operation?

The loaded coal pile on a scraper conveyor can dampen vibration to some extent, and the causes of this dampening effect and its effects are as follows:

1. Reducing impact and wear: The loading coal pile can reduce the impact force on the chain drive system. As the coal pile can absorb and cushion the impact energy of lump coal, it can reduce the wear of chain rings, scrapers and central groove and prolong their service life.

2. Mass damping effect: The loaded coal pile, as a vibration damping body with a certain mass and rigidity, can attenuate the amplitude and frequency of vibration through the mass damping effect. The mass and rigidity of the coal pile will produce a reaction force on the vibration system, thus suppressing the vibration of the chain drive system.

3. Dynamic stability: The loaded coal pile can provide a stable load and support for the chain drive system, and improving its dynamic stability, which balances and disperses shock forces in the drive system and reduces the effects of vibration transfer to the chain and other components.

4. Reduction of chain runout: The loaded coal pile can effectively reduce chain runout. Chains are susceptible to runout due to load impacts during transmission, which generates vibration and noise, while loaded coal piles are usually located above or to the side of the chain, it can provide additional restraint and stability, thus suppressing the runout of the chain.

- Practical Significance: How can the findings of your study be practically applied to enhance the stable operation and structure optimization of scraper conveyors in mining operations?

Applying the results of this study to mining operations to improve the stable operation and structural optimization of scraper conveyors, I think we can start from the following aspects:

1. Optimization of design parameters: The design parameters of the scraper conveyor can be optimized based on the results of the study, For example, the rigidity and stability of the chain drive system can be improved and the generation of vibration and noise can be reduced by reasonably selecting the size, material and geometry of the chain, scraper and central groove.

2.Installation of vibration control devices: According to the research results, it can be considered to introduce appropriate vibration control devices on the scraper conveyor, such as vibration dampers, vibration-damping shims and so on. These devices can reduce the transmission and diffusion of vibration, improve the stability of the system, and reduce the impact of vibration on the scraper conveyor.

3. Regular inspection and maintenance: according to the results of the study, it is possible to formulate a regular inspection and maintenance plan for the scraper conveyor, such as regular inspection of chain tension, wear and looseness, and timely adjustment and replacement of the chain, which is able to prolong the service life of the chain and scraper conveyor.

Overall, applying the results of the study to mining operations can be achieved by optimizing the design parameters, vibration control, and regular inspection and maintenance, etc. By applying these measures in a comprehensive manner, the stable operation of the scraper conveyor can be ensured, and more efficient, reliable and safe mining operations can be achieved.

- Future Research: Are there any specific areas of future research that you believe should be explored based on the results of your study or any limitations you encountered?

The present study is limited by the conditions of the test bench, the values of mass and height of impact load are conservative, and the effect of terrain change on the dynamics of the scraper conveyor under impact conditions is not studied. Future research will further improve the impact test bench of scraper conveyor by using a larger mass of impact load and a higher impact height for impact test, it can more realistically simulate the real working conditions and improve the reliability and accuracy of the study. At the same time, it is necessary to add the test conditions of terrain change, to study the influence of factors such as surface undulation on the dynamic characteristics of the scraper conveyor under impact conditions, to further improve the research on the dynamic characteristics of the scraper conveyor under impact conditions.

- Industry Impact: How might the mining industry benefit from a better understanding of the dynamic characteristics of scraper conveyor chain drive systems, particularly in terms of safety and efficiency?

A better understanding of the dynamic characteristics of the scraper conveyor chain drive system, particularly in terms of safety and efficiency, will bring the following benefits to the mining industry :

1. Improve efficiency: Understanding the dynamic characteristics of a scraper conveyor chain drive system can help optimize equipment operation and improve production efficiency. By knowing exactly how the chain drive system works and responds, you can optimize the design and operation of the system while reducing energy consumption and production downtime.

2. Enhanced equipment safety: A good understanding of the dynamic characteristics of chain drive systems can help to predict and prevent potential failures, it is possible to reduce the risk of damage to equipment and enhance safety in the workplace by taking preventive and maintenance measures.

3. Reducing maintenance costs: Understanding the dynamic characteristics of the chain drive system can help detect and diagnose potential problems with the scraper conveyor and take timely maintenance measures to reduce the occurrence of system failures and damage. This can reduce equipment maintenance costs and losses due to production downtime.

4. Extend the life of the equipment: through a deeper understanding of the dynamic characteristics of the chain drive system, it is possible to extend the life of the equipment by optimizing the structure of the equipment in order to reduce the degree of wear and fatigue of the equipment.

In conclusion, a better understanding of the dynamic characteristics of the chain drive system of a scraper conveyor has important benefits for the mining industry. It not only helps to improve the safety and efficiency of the equipment and prolong the equipment life, but also reduces maintenance costs and energy consumption, and improves the stability and reliability of the production system. This will promote the sustainable development and competitiveness of the entire mining industry.

- Research Contribution: What do you consider the main contribution of your research to the field of mining equipment and conveyor systems?

There are many shortcomings in this study at this time, but with the theoretical results obtained in this study, I think this study can make the following contributions to the field of mining equipment and conveyor systems:

1. Improve the safety performance of scraper conveyor: Studying the dynamic characteristics of scraper conveyor under impact conditions can help to identify the vulnerable points and safety hazards of the scraper conveyor, and determine the location where the scraper conveyor fails first under impact conditions, so as to improve its safety performance.

2. Optimize the design of scraper conveyor: Studying the dynamic characteristics of the scraper conveyor under impact conditions can help to understand the working principle and law of the scraper conveyor, so as to provide a basis for its design and improvement. By optimizing the design, the efficiency and reliability of the scraper conveyor can be improved, and energy consumption and maintenance costs can be reduced.

3. Reduce scraper conveyor maintenance costs and improve its service life: Studying the dynamic characteristics of the scraper conveyor under impact conditions can help reduce scraper conveyor maintenance costs and improve its service life by optimizing the structure of the scraper conveyor to reduce its wear and tear.

 

To Reviewer 2:

Comment 1. Revisit the paper to correct some typos and minor grammatical mistakes.

I have revisited the paper based on your comments and corrected some typos and minor grammatical mistakes, thank you very much for your valuable comments.

Comment 2. Abstract: Highlight the key research results and implications.

The abstract section has been revised with reference to your comments to highlight the key research results and implications and has been highlighted in blue.

Comment 3. Ensure proper spacing between wordings and symbols throughout the paper. For example, correct “development[1].”, “equipment[2].”…

The incorrect section has been corrected based on your comments to ensure proper spacing between wordings and symbols throughout the paper.

Comment 4. Section 1 Introduction:

(a) Update the list of references to cover more recent 2-year published articles. Elaborate on the results and limitations of existing works.

(b) Summarize the research contributions of the paper.

(a) The list of reference has been updated in response to your comments and has been highlighted in blue., and the articles cited is now published in 2020 or later, and the results and limitations of existing works have been elaborated.

(b) The research contributions of the paper have been summarized in response to your comments and have been highlighted in blue.

Comment 5. Section 2 Theoretical basis:

(a) Clarify if the formulation follows existing works as background information. If yes, include references.

(b) Ensure the presented content helps readers understand the rest of the technical details in the coming sections.

(a) The theoretical content of this section is based on existing works as background information, and the works referenced are:

[1] Sakino K, Tomii M. Hysteretic behavior of concrete filled square steel tubular beam-columns failed in flexure. Transactions of the Japan Concrete Institute. 1981;3(6):439-446.

[2] Zheng A X, Wu L. Research on numerical of crack growth of gravity dam based on XFEM. China Rural Water and Hydropower. 2011;03:109-112.

(b) This section introduces the theory of impact dynamics and the basic equations derived based on it to solve the impact problem, which is helpful for the design of the impact test scheme of the scraper conveyor and it can provide theoretical basis for the simulation study of the scraper conveyor impact problem in the future.

Comment 6. Section 3 Research Program:

(a) Add an introductory paragraph before Subsection 3.1.

(b) Justify if the models of scraper conveyors and sensors are common in the industry.

(c) The test schemes only covered limited settings. Justify if they align with the ranges in reality.

(a) An introductory paragraph has been added before subsection 3.1 in response to your comments and has been highlighted in blue.

(b) The models of scraper conveyors and sensors are common in the industry and which able to accurately simulate realistic working conditions. This explanation has also been added to the introductory paragraph above.

(c) The range of variables set in the test scheme is the most consistent with the actual working conditions that can be simulated by this test bench, which can well simulate the realistic working conditions. This explanation has also been added to the introductory paragraph above.

Comment 7. Section 4 Test results and analysis:

(a) Add an introductory paragraph before Subsection 4.1.

(b) Enhance the resolutions of all figures. Ensure no content is blurred.

(c) If there are subfigures, ensure the captions are included for all subfigures.

(d) Table 5, apart from average values, includes more information.

(e) Comparison between proposed work and existing works is expected.

(a) An introductory paragraph has been added before subsection 4.1 in response to your comments and has been highlighted in blue.

(b) The resolution of all figures has been increased based on your comments to ensure as much blur-free content as possible.

(c) This paper includes a number of subfigures, all of which have been ensured to have captions.

(d) The data of standard deviation have been added to Table 5 based on your comments and has been highlighted in blue.

(e) Because the simulation of the impact condition of the scraper conveyor is still in progress due to the problem of parameter setting, I am very sorry that I cannot provide you with the comparison between proposed work and existing works before the paper revision deadline, but if possible I 'll provide you with this comparison after the simulation is complete.

Comment 8. Elaborate on future research directions.

Future research directions have been elaborated in more detail based on your comments and have been highlighted in blue in section 5, "Conclusions".

---

## [Decision Letter · Decision Letter 1]

5 Feb 2024

Dynamic characteristics of scraper conveyor chain drive system under the impact condition of lump coal

PONE-D-23-28370R1

Dear Dr. Zeng,

We’re pleased to inform you that your manuscript has been judged scientifically suitable for publication and will be formally accepted for publication once it meets all outstanding technical requirements.

Kind regards,

Hugh Cowley

Staff Editor

PLOS ONE

Additional Editor Comments (optional):

Reviewers' comments:

Reviewer's Responses to Questions

**Comments to the Author**

1. If the authors have adequately addressed your comments raised in a previous round of review and you feel that this manuscript is now acceptable for publication, you may indicate that here to bypass the “Comments to the Author” section, enter your conflict of interest statement in the “Confidential to Editor” section, and submit your "Accept" recommendation.

Reviewer #1: All comments have been addressed

Reviewer #2: All comments have been addressed

2. Is the manuscript technically sound, and do the data support the conclusions?

Reviewer #1: Yes

Reviewer #2: Yes

3. Has the statistical analysis been performed appropriately and rigorously? 

Reviewer #1: Yes

Reviewer #2: N/A

4. Have the authors made all data underlying the findings in their manuscript fully available?

Reviewer #1: Yes

Reviewer #2: No

5. Is the manuscript presented in an intelligible fashion and written in standard English?

Reviewer #1: No

Reviewer #2: Yes

6. Review Comments to the Author

Reviewer #1: All the suggested changes are incorporated well. This paper can be considered now for publication.

Reviewer #2: All of my comments are addressed. The quality of the paper is significantly enhanced. I recommend accepting the paper for publication.

7. PLOS authors have the option to publish the peer review history of their article (what does this mean?). If published, this will include your full peer review and any attached files.

Reviewer #1: No

Reviewer #2: **Yes: **Kwok Tai CHUI

---

## [Editor Report · Acceptance letter]

21 Feb 2024

PONE-D-23-28370R1 

PLOS ONE

Dear Dr. Zeng, 

I'm pleased to inform you that your manuscript has been deemed suitable for publication in PLOS ONE. Congratulations! Your manuscript is now being handed over to our production team.

Kind regards, 

on behalf of

Mr Hugh Cowley 

Staff Editor

PLOS ONE